# A study of synthetic $^{13}CH_4$ retrievals from TROPOMI and Sentinel 5/UVNS

Edward Malina[1], Haili Hu[2], Jochen Landgraf[2], and Ben Veihelmann[1]

[1]Earth and Mission Science Division, ESA/ESTEC, Keplerlaan 1, Noordwijk, the Netherlands.
[2]SRON Netherlands Institute for Space Research, Utrecht, the Netherlands.

**Correspondence:** Edward Malina (edward.malina.13@alumni.ucl.ac.uk)

**Abstract.** Retrievals of methane isotopologues have the potential to differentiate between natural and anthropogenic methane sources types, which can provide much needed information about the current global methane budget. We investigate the feasibility of retrieving the second most abundant isotopologue of atmospheric methane ($^{13}CH_4$, roughly 1.1% of total atmospheric methane) from the Shortwave Infrared (SWIR) channels of the future Sentinel 5/UVNS and current Copernicus Sentinel 5 Precursor TROPOMI instruments. With the intended goal of calculating the $\delta^{13}C$ value, we assume that a $\delta^{13}C$ uncertainty of better than 1‰ is sufficient to differentiate between source types, which corresponds to a $^{13}CH_4$ uncertainty of <0.02 ppb. Using the well established Information Content analysis techniques and assuming clear sky, non-scattering conditions, we find that the SWIR3 (2305 - 2385 nm) channel on the TROPOMI instrument can achieve a mean uncertainty of <1 ppb, while the SWIR1 channel (1590 - 1675 nm) on the Sentinel 5 UVNS instrument can achieve <0.68 ppb, or <0.2 ppb in high SNR cases. These uncertainties combined with significant spatial and/or temporal averaging techniques can reduce $\delta^{13}C$ uncertainty to the target magnitude or better. However, we find that $^{13}CH_4$ retrievals are highly sensitive to errors in a priori knowledge of temperature and pressure, and accurate knowledge of these profiles are required before $^{13}CH_4$ retrievals can be performed on TROPOMI and future Sentinel 5/UVNS data. In addition, we assess the assumption that scattering induced light path errors are cancelled out by comparing the $\delta^{13}C$ values calculated for non-scattering and scattering scenarios. We find that there is a minor bias in $\delta^{13}C$ values from scattering and non-scattering retrievals, but this is unrelated to scattering induced errors.

## 1 Introduction

With the recent launch of the TROPOspheric Monitoring Instrument (TROPOMI) aboard the Copernicus Sentinel 5 Precursor (S5P) satellite, global monitoring of methane concentrations and fluxes has been put firmly at the forefront of the efforts towards understanding global Greenhouse Gas (GHG) emissions and climate change. Methane, while present in much smaller concentrations than the main anthropologically influenced GHG carbon dioxide ($CO_2$), has stronger global warming potential than $CO_2$ (IPCC, 2014). Crucially methane is less understood, with bottom up estimations (observations from in situ sites/inventory compilations) showing poor agreement with top down estimates (resulting from measurements assimilated into chemistry transport models (CTMs)). This disagreement is likely due to currently limited observations, incorrect atmospheric transport assumptions, or uncertainties associated with bottom up inventories and uncertainties in modelling $CH_4$ chemical

losses (Kirschke et al., 2013). This is best shown through the current multiple, sometimes contradicting theories as to the reasons for the pause in atmospheric methane growth at the start of the last decade, and its subsequent rise several years later (Kai et al., 2011; Aydin et al., 2011; Nisbet et al., 2014, 2016; Mcnorton et al., 2016; Rigby et al., 2017).

Towards this end, it is necessary to build up a greater understanding of global methane sources and sinks in order to allow for better predictions on how the climate will be affected, and to develop potential mitigation strategies. Numerous satellite missions have been launched in order to provide this understanding, starting with the SCanning Imaging Absorption spectroMeter for Atmospheric CartograpHY (SCIAMACHY) aboard ENVISAT (Bovensmann et al., 1999) launched in 2002 (ceasing operations in 2012), and continuing with the Greenhouse gases Observing SATellite (GOSAT; (Kuze et al., 2009)), launched in 2009 (currently operational). Both SCIAMACHY and GOSAT have broken significant ground in relation to providing global and regional estimates of methane concentrations (Frankenberg et al., 2008; Kort et al., 2014; Yoshida et al., 2011; Schepers et al., 2012; Parker et al., 2016; Buchwitz et al., 2017), but both instruments retain drawbacks that prevent the closing of the gap between the top down and bottom up estimates of global methane. Firstly, SCIAMACHY stopped functioning in 2012 and cannot provide any new data. Secondly SCIAMACHY is identified to have a single sounding precision of between 30-80 ppb for methane retrieval (not discriminating between isotopologues), indicating that all SCIAMACHY retrievals require large temporal and/or spatial averaging in order to provide high certainty methane volume mixing ratio estimations (Kort et al., 2014), likely making identifying localised high frequency fluxes impossible e.g. Buchwitz et al. (2017). Such fine scale observations are required in order to improve top down methane estimates. GOSAT-Thermal and Near Infrared Sensor for Carbon Observation - Fourier Transform Spectrometer (TANSO-FTS) has higher sensitivity and spatial resolution than SCIAMACHY, but has low spatial sampling (Kuze et al., 2012). S5P TROPOMI and the future Sentinel 5 (S5)/Ultra-Violet, Visible, Near Infrared, Shortwave Infrared (UVNS) (Ingmann et al., 2012) instruments aim to build upon the legacy of SCIAMACHY by providing methane measurements at higher precision, higher spatial resolution and near daily global coverage. The goals and capabilities of the TROPOMI methane product are described in more detail in (Hu et al., 2016), and the S5/UVNS methane product goals are outlined in Ingmann et al. (2012).

Nisbet et al. (2016) states that measurements or retrievals of methane currently do not provide sufficient information in order to definitively define the methane budget, since such measurements do not include any information on the source type or contribution. This is highlighted by Kirschke et al. (2013) and Saunois et al. (2017) who show significant uncertainty in the global methane budget due to the often unknown or poorly understood contribution of individual source types, especially wetlands. These studies make it clear that in order to understand the global methane budget, it is important to understand the nature of the emissions (i.e. whether they are biogenic or abiogenic). It has been shown that methane source types can be differentiated through the use of the ratio of the two most common methane isotopologues, $^{12}CH_4$ (comprising $\sim$98% of atmospheric methane) and $^{13}CH_4$, (making up $\sim$1.1% of atmospheric methane), typically through a ratio of these isotopologues known as the $\delta^{13}C$ ratio. The global variability of this ratio has often been used in studies relating to understanding the global methane budget (Nisbet et al., 2016; Schaefer et al., 2016; Rigby et al., 2017), and global shifts in this ratio have even been touted as one of the possible main reasons for the recent growth of global methane. However knowledge of this ratio is severely limited, and typically based on a small number of flask air samples (Nisbet et al., 2016; Rigby et al., 2017), or from very specific

field campaigns (Rella et al., 2015; Fisher et al., 2017). Recently, interest in expanding global knowledge of the $\delta^{13}C$ ratio through the use of satellite measurements has been increasing, firstly through limb measurements with the SCISAT ACE-FTS instrument (Buzan et al., 2016), and also through investigations of potential future instruments (Weidmann et al., 2017; Malina et al., 2018). Buzan's results are important because they represent the first attempt at calculating the $\delta^{13}C$ measurement from a satellite instrument. However, Buzan et al. (2016) are unable to draw any conclusions from their results, due to poor agreement with CTMs and in situ balloon measurements, which is largely explained through errors in spectroscopy.

In this study we investigate the possibility of retrieving $\delta^{13}C$ with the recently launched TROPOMI instrument, and the future Sentinel 5/UVNS instrument, focusing on synthetic measurements using the well established Information Content (IC) analysis techniques introduced by Rodgers (2000). The TROPOMI and UVNS instruments are based on different technology than used previously for methane isotopologue measurements. Both ACE-FTS and GOSAT-TANSO-FTS are high spectral resolution FTSs, (e.g. 0.02 cm$^{-1}$ for ACE-FTS and 0.2 cm$^{-1}$ for GOSAT), while TROPOMI and UVNS are push broom spectrometers and have a lower spectral resolution (0.45 cm$^{-1}$). However TROPOMI and UVNS are expected to be able to capture measurements at higher SNR, and therefore the key question becomes whether SNR or spectral resolution is the key limiting factor in the retrieval of methane isotopologues. TROPOMI and UVNS share a Shortwave Infrared (SWIR) spectral band known as SWIR3, covering the range 2305-2385 nm, while UVNS also includes an additional SWIR band known as SWIR1, covering the range 1590-1675 nm. The IC analysis techniques identified above will be used on both of these bands in this paper.

In this paper, we primarily focus on retrievals made under the assumption that all atmospheric scattering effects are cancelled out. This is based on the methods of Frankenberg et al. (2005, 2011); Parker et al. (2011); Schepers et al. (2012), where the ratio of two spectrally close trace gases are taken in order to remove scattering artefacts. Therefore the majority of simulations are performed assuming clear sky conditions and all scattering is turned off in the forward model. The assumes that light path modifications due to atmospheric scattering affect spectrally close species in a similar fashion. Previous applications of this assumption use the strong absorbers methane and carbon dioxide, in this work when calculating the $\delta^{13}C$ metric using non-scattered retrievals of $^{12}CH_4$ and $^{13}CH_4$, it is assumed that all scattering effects are cancelled out since the two isotopologues can be considered as separate species that are spectrally very close, and therefore all common spectral artefacts will be minimised. Because $^{13}CH_4$ is a weak absorber there is an argument that scattering may affect $^{13}CH_4$ and $^{12}CH_4$ differently. Therefore we have included a section where we retrieve the isotopologue VMRs under scattering conditions, and compare the results with non-scattering retrievals.

This paper is structured as follows:

Section 2 describes the instruments under consideration, the tools and models used to simulate these instruments and perform all relevant analyses, and the metrics used to assess the model outputs. Section 3 presents a detailed information content analysis focusing on the SWIR1 band present in S5/UVNS, but not in S5P/TROPOMI. Section 4 is as Sect 3, but focusing on the SWIR3 band present in both S5P/TROPOMI and S5/UVNS Section 5 is as Sects 3 and 4, but is focused on a dual band retrieval from both SWIR channels in Sentinel 5. Section 6 shows a comparison between scattering and non-scattering retrievals for all bands. Section 7 presents a brief discussion of the methods used in this research and conclusions are drawn in Sect 8.

## 2 Study setup, requirements, models and instruments

### 2.1 TROPOMI and Sentinel 5

S5P TROPOMI (Veefkind et al., 2012) was successfully launched into low earth orbit (LEO) on the 13th of October 2017, with the aim to provide global information on air quality, climate and the ozone layer. The key products that are to be published from TROPOMI include, $O_3$, $SO_2$, $NO_2$, CO, $CH_4$, $CH_2O$ and aerosol properties. These trace gas products are measured through solar backscatter in four separate spectral ranges, ultra violet (UV), visible (VIS), near infrared (NIR) and SWIR, which are described in more detail in Table 1 below. The TROPOMI instrument is built upon the heritage of previous missions aimed at studying the products mentioned earlier, namely the Global Ozone Monitoring Experiment (GOME; (Burrows et al., 1997), SCIAMACHY (Bovensmann et al., 1999), the Ozone Monitoring Instrument (OMI; (Levelt et al., 2006)), and GOME-2 (Callies et al., 2000). TROPOMI provides a significant advance in instrument technology over SCIAMACHY, with finer spatial resolution (7.5 x 7.5 km vs 30 x 240 km) and measurement uncertainty. The first results from TROPOMI are starting to be published (Borsdorff et al., 2018; Hu et al., 2018) and are already providing significant new results to the community.

Sentinel 5 (Pérez Albiñana et al., 2017) due for launch in 2022 on the MetOp-Second Generation (SG)-A satellite, will compliment the results of TROPOMI, providing global information on GHGs and pollutants at high spatial resolution. MetOp-SG-A is the first of a pair of satellites that are designed to complement each other, but carry different instruments unlike the current MetOp satellites. The MetOp-SG series of satellites will eventually comprise of 6 separate satellites, each with an 8.5 year lifetime. Sentinel 5/UVNS is very similar to TROPOMI, with both missions having similar instrument types, orbit altitudes, but differing descending nodes (S5P - 13:30 and S5 - 09:30) such that the instruments will capture measurements under differing solar zenith angles. The key differences are the minor variations in the spectral bands and the inclusion of the SWIR1 band, which allows for the retrieval of $CO_2$, and multiple band retrievals of $CH_4$; in the UV/VIS range CHOCHO will be an additional product of Sentinel 5/UVNS not present in the TROPOMI retrieval products. The spectral bands and spectral resolutions of S5P/TROPOMI and S5/UVNS are described in Tables 1 and 2 below.

**Table 1.** Characteristics of S5P/TROPOMI spectral bands

| Band | UV1 | UV2 | UVIS | VIS | NIR1 | NIR2 | SWIR3 |
|---|---|---|---|---|---|---|---|
| Spectral range | 270-300 nm | 300-320 nm | 310-405 nm | 405-500 nm | 675-725 nm | 725-775 nm | 2305-2385 nm |
| Spectral resolution | 1.0 nm | 0.5 nm | 0.55 nm | 0.55 nm | 0.5 nm | 0.5 nm | 0.25 nm |

**Table 2.** Characteristics of S5/UVNS spectral bands

| Band | UV1 | UV2VIS | NIR1 | NIR2a | NIR2 | SWIR1 | SWIR3 |
|---|---|---|---|---|---|---|---|
| Spectral range | 270-310 nm | 300-500 nm | 685-710 nm | 745-755 nm | 755-773 nm | 1590-1675 nm | 2305-2385 nm |
| Spectral resolution | 1.0 nm | 0.5 nm | 0.4 nm | 0.4 nm | 0.4 nm | 0.25 nm | 0.25 nm |

## 2.2 RemoTeC

In this work we apply the well established RemoTeC retrieval software designed for TROPOMI (Butz et al., 2010; Hu et al., 2016, 2018); RemoTeC is a solar backscatter model based around the radiative transfer model developed by Hasekamp and Landgraf (2002). RemoTeC uses a 36-layer plane-parallel atmosphere, and including multiple atmospheric scattering effects and surface reflection physics. RemoTeC is fully described in Butz et al. (2012); Hu et al. (2016), and we refer to these papers for full details about the software. However, we briefly summarise the algorithm here.

RemoTeC infers an atmospheric state vector from spectral measurements in the NIR band (757-774 nm) and the SWIR bands (1590-1675 nm, 2305-2385 nm), via the inversion method known as the Philips-Tikhonov regularisation scheme (Tikhonov, 1963; Phillips, 1962). This scheme is required because the spectral measurements typically do not contain enough information to retrieve all state vector elements independently. Using a non-linear iterative scheme, the state vector is estimated by minimising the following cost function.

$$\hat{\mathbf{x}} = min(\|\mathbf{S_y}^{1/2}(\mathbf{F}(\mathbf{x}) - \mathbf{y})\|^2 + \gamma\|\mathbf{W}(\mathbf{x} - \mathbf{x_a})\|^2), \tag{1}$$

where $\mathbf{S}_y$ is the measurement error covariance matrix, $x$ and $x_a$ are the state and priori state vectors. $F(x)$ is the forward model, representing the physical model of the atmosphere. $y$ is the vector containing the measurement. $W$ is a diagonal weighting matrix rending the side constraint dimensionless, and ensuring that only relevant parameters contribute to the norm. $\gamma$ is the regularisation parameter that is determined through the L curve method (Hansen, 2000). The regularisation parameter can be modified manually if deemed appropriate.

The state vector $x$ is composed of the following elements:

- CH$_4$ in 12 vertical layers, a priori values from the CTM TM5.

- CO total column, a priori values from the CTM TM5.

- H$_2$O total column, a priori calculated from ECMWF humidity.

- Aerosol column, a priori calculated from aerosol optical thickness = 0.1 at 760 nm.

- Aerosol size parameter, a priori fixed value.

- Aerosol height parameter, a priori fixed value.

- Lambertian surface albedo in the NIR band, a priori maximum of measured reflectance in the NIR.

- First-order spectral dependence of surface albedo in the NIR band, a priori fixed value.

- Lambertian surface albedo in the SWIR band, a priori maximum of measured reflectance in the SWIR.

- First-order spectral dependence of surface albedo in the SWIR band, a priori fixed value.

- Spectral shift NIR, a priori fixed value.

- Spectral shift SWIR, a priori fixed value.

- Fluorescence emission at 755 nm, a priori fixed value.

- Fluorescence spectral slope, a priori fixed value.

In order to apply the software to methane isotopologues, some minor changes were required, primarily in the spectroscopy, which we describe here, but also the state vector, where $^{13}CH_4$ and $^{12}CH_4$ replace purely $CH_4$. RemoTeC primarily draws its spectroscopic data from HITRAN2008 (Rothman et al., 2009), amongst others. However these databases were found to be deficient in methane isotopologue spectral lines (mainly $^{13}CH_4$), therefore the spectroscopy was updated to HITRAN2012 (Rothman et al., 2013). The HITRAN2012 database includes line parameters for all isotopologues of the same molecule

assuming fixed abundance ratios, such that the total $CH_4$ absorption cross-section can be computed conveniently based on the total atmospheric profile (i.e. the sum over all isotopologues). In the case of $^{13}CH_4$ this scaling is done through a multiplication factor of 0.0111. In the context of this performance study, we can justify keeping this scaling factor since we aim to describe the feasibility of methane isotopologue retrieval, and not perform retrievals from current TROPOMI data. All other aspects of the RemoTeC algorithm (state vector parameters, etc) are as identified in Hu et al. (2016).

Note that the current version of RemoTeC is optimised for the SWIR3 band of TROPOMI and not Sentinel 5/UVNS, and some modifications were made in order to apply RemoTeC to Sentinel 5 simulations. In addition, because RemoTeC has heritage with GOSAT, the additional SWIR1 channel can also be used. This study uses the SWIR3 TROPOMI noise model applied in Hu et al. (2016), in addition we employ a noise model that is representative for UVNS.

## 2.3   Synthetic study: the global ensemble

We decided in this study to focus on synthetic measurements since TROPOMI is still in an early mission phase, and methane isotopologues are still an unexploited area. S5/UVNS will not be launched until 2022 and therefore data will not be available until then. Further because this is a feasibility study, full control of synthetic scenarios along with the known "truth" for verification is a significant benefit.

The synthetic data in this study are effectively the same as those outlined by Butz et al. (2012); Hu et al. (2016), and are

described in detail by Hu et al. (2016), though note that in this paper only the SWIR spectra are considered. To summarise, the synthetic database comprises a wide range of realistic conditions that TROPOMI and UVNS will/are be expected to encounter, such as surface types, atmospheric conditions, solar zenith angles, nadir viewing and orbit. The measurements are designed to simulate the four main seasons over the course of a year, and include examples of aerosols and cirrus clouds. All measurements in the databases are derived from a of combination chemistry transport models (TM5, ECHAM5-HAM) and satellite products

(MODIS and SCIAMACHY). The synthetic data are sampled over the globe on a 2.79° x 2.8125° latitude by longitude grid, with only land surfaces considered.

Aerosol parameters are derived from a sophisticated combination of the ECHAM5-HAM (Stier et al., 2005) global aerosol model and MODIS observations. The ECHAM-HAM model is run at the same spatial resolution and in vertical layers up to the mid-stratosphere, and contains five different chemical species on a superposition of seven log-normal size distributions. The Aerosol Optical Depth (AOD) is scaled to match observations from the MODIS satellite instrument in 2007, all of which is considered in the synthetic spectra generation (Butz et al., 2012; Hu et al., 2016). In contrast, the aerosol representation in the retrieval scheme is relatively simple, where only one aerosol type is considered for which the aerosol column amount, aerosol size parameter and aerosol height parameter are derived. Both Rayleigh and Mie scattering are considered in RemoTeC, where only spherical particulates are considered. The result of these differences in the treatment of aerosols between the synthetic database and the retrieval algorithm is the deliberate introduction of systematic errors into the retrieval process through the forward model. This is similar to the spectral sampling approach, where the synthetic spectra are calculated using line by line parameters, and the retrieval algorithm using a correlated k-means method.

Using these simulated scenarios, and the LINTRAN forward model (Hasekamp and Landgraf, 2002) we generate 11041 simulated synthetic spectra. In order to include simulated TROPOMI/Sentinel 5 instrument effects the synthetic spectra are convolved with a Gaussian instrument line shape function (ILSF) with a full width half maximum (FWHM) of 0.25 nm. The instrument noise models are as described in Hu et al. (2016) for the SWIR3 bands on TROPOMI and UVNS, while the noise model for the SWIR1 band on UVNS is based on characterisation work performed at ESA. Both of these noise models include shot noise and inherent instrument noise terms.

In essence the RemoTeC software is comprised of two distinct elements; the first element is a forward model which takes in the synthetic database of atmospheric profiles and surface conditions, and converts these into top of atmosphere radiances (including aerosol and surface albedo effects), and includes instrument ILSF and noise effects. The second element is the retrieval algorithm which then retrieves the trace quantities back from the simulated spectra, and is based on the Philips-Tikhonov regularisation scheme. The retrieval forward model allows introduction of deliberate inconsistencies with the synthetic forward model, in order to simulate forward model errors. For example the synthetic scenario spectra are generated using line by line spectroscopy, while the retrieval forward model uses the linear k-method (only applicable to scattering retrievals (Hasekamp and Landgraf, 2002)) as an approximate spectral sample technique, which is quicker than the line by line method. Errors in the spectroscopy are not modelled in this study.

Through this paper, we will refer to "latitudinal bands", which we split in three distinct areas. Tropical ( 0-20°), mid-latitude ( 20-60°) and high latitude (>60°). These are typically how model atmospheres are split (e.g. mid-latitude summer etc). Surface conditions will cause the results in this bands to vary, and we identify any regions that show significant deviation.

## 2.4 Study requirements

Fundamentally, the goal of methane isotopologue retrieval is to differentiate between methane source types. To achieve this we calculate the $\delta^{13}C$ value, which is the currently accepted metric used for this differentiation (Rigby et al., 2012; Schaefer et al., 2016). Nisbet et al. (2016) identify that for a given source type $\delta^{13}C$ values typically vary by up to 1‰ over the course of a year, which means that TROPOMI/Sentinel 5/UVNS need to achieve 1‰ total uncertainty or better (<0.1‰, if seasonal

variations are to be observed (Nisbet et al., 2016). However Buzan et al. (2016), Weidmann et al. (2017) and Malina et al. (2018) identify that with current satellite retrieval techniques, this level of precision is difficult, since this would require total $^{13}CH_4$ column errors <0.02 ppb, which equates to roughly 0.1% $^{13}CH_4$ total column error. Assuming that total column $^{13}CH_4$ VMR is roughly 0.011% of a total column VMR of $CH_4$ based on the HITRAN apportionment.

This is not currently possible for instantaneous measurements even for higher concentration species. However, if the random error is sufficiently low, higher order precisions could be obtainable with spatio/temporal averaging given the high repeat cycles of S5P/TROPOMI and S5/UVNS (planned). The question then becomes, what may be technically possible with current satellite instruments, and how such data can be leveraged. Malina et al. (2018) identify a target total uncertainty for $\delta^{13}C$ of 10‰ as a more realistic and potentially achievable value (based on simulations with GOSAT-2). However, further analysis (e.g. Fisher

et al. (2017)) and discussions suggest that although such variations in $\delta^{13}C$ between sources may be possible when measured at the surface, it is unlikely to remain true when viewed from space. For this study we are aiming for a $\delta^{13}C$ uncertainty of 1‰, as of possible benefit to the wider community. Rigby et al. (2017); Nisbet et al. (2016) indicate that the $\delta^{13}C$ signature of sources can vary by 1‰ over the course of the year, and given S5P and S5 are envisaged to operate for several years, we will aim for this target uncertainty.

**2.5   Study structure**

The primary aim of this study is to establish the IC (Rodgers, 2000) of $^{13}CH_4$ in simulated TROPOMI and UVNS retrievals, similar to the study by Malina et al. (2018). Malina et al. (2018) based their study on an optimal estimation routine (Rodgers, 2000), and experimented with a priori covariance matrices for $^{13}CH_4$. This paper builds on Malina et al. (2018), but is significantly different to Malina et al's work, since we are investigating different satellite instruments, in addition to more advanced

atmospheric scenarios and scenes. Another fundamental difference between the studies is that RemoTeC is based on the Philips-Tikhonov regularisation scheme, and therefore experimenting with a priori covariance matrices is no longer necessary. In theory there should be no difference in the results from using the two different methods, but in practise care must be taken to ensure that the algorithms are fully optimised for minor species.

     Based on the methods of Hu et al. (2016); Malina et al. (2018); Rodgers (2000) we use the following metrics to identify the

25 IC of $^{13}CH_4$ from TROPOMI:

     – Column averaging kernels: Indicating sensitivity of the retrieved state vector to the truth.

     – Degrees of Freedom of Signal (DFS): Measure for the number of pieces of information in a retrieval that can be associated with the state vector. Defined by the trace of the full averaging kernel.

     – Total column errors: Indicating the precision and accuracy of retrievals. In this synthetic study the errors are defined as

the difference between the synthetic "truth" and the retrieved quantity. Therefore all errors include both precision and systematic errors.

     – Fit quality: The ($\chi^2$) test is used, outlining quality of retrieval fits (as (Galli et al., 2012)).

– Jacobians: Sensitivity of the forward model to state vector changes. The Jacobians are defined as the sensitivity of the forward model to changes in the state vector. In this study we investigate how the total column Jacobians vary between the isotopologues, however Malina et al. (2018) give examples of how the Jacobians vary on a profile basis.

These metrics are calculated for the $^{13}CH_4$ retrievals using the SWIR3 band (TROPOMI and UVNS), SWIR1 (UVNS) and a combination of SWIR1 and SWIR3 bands (UVNS), under the assumption that retrievals for $^{12}CH_4$ will exhibit similar values to those shown in Hu et al. (2016).

Following this, we investigate the sensitivity of $^{13}CH_4$ and $\delta^{13}C$ retrievals to prior knowledge of the atmospheric state focusing on the following areas:

– A priori methane profile: Ideally, the retrieval will be insensitive to the choice of a priori methane profiles. To test this assumption, we investigated the effects of perturbation ($\pm2\%$) of the a priori profile, which is otherwise set to the synthetic 'truth' in this study.

– A priori water vapour profile: In the same way as methane, we investigate the effects of imprecise knowledge of the water vapour column ($\pm10\%$), since water vapour exhibits strong absorption features in this spectral range, it can interfere with methane retrievals especially in the case of focusing on a weak absorber such as $^{13}CH_4$.

– Pressure: Here we introduce a $\pm0.3\%$ error into the a priori pressure profile. Pressure errors can affect the retrieval of methane in two ways, the first is through the retrieved air column which converts the total column concentration of methane into volume mixing ratios (VMRs). The second is through pressure dependence of the spectroscopy cross sections.

– Temperature: Errors in the temperature profile are introduced through the temperature dependence of the spectroscopic cross sections ($\pm2$ K).

The magnitudes of the errors used in the prior knowledge are based on the errors derived by Hu et al. (2016) and Landgraf et al. (2016) from the CTMs used to provide the prior atmospheric data. For methane TM5 was used, all of the other data are based on ECMWF. Note that the magnitudes in this study are worst case scenarios, and therefore the bias errors indicated in this section will be the maximum. Hu et al. (2016) do not indicate any significant non-linear behaviour in systematic error investigations, suggesting that different magnitude errors in the a priori methane profile will yield similar systematic errors. Typical standard deviations of the CTMs were found to be significantly lower (Landgraf et al., 2016). In addition, when calculating the errors induced in the $\delta^{13}C$ ratio, errors from calculating the $^{12}CH_4$ VMR are included, since the methane a priori profile is used for both $^{13}CH_4$ and $^{12}CH_4$.

In addition to the atmospheric state, we investigate the following instrument/calibration errors:

– Radiometric offset (additive): A spectrally constant offset ($\pm0.1\%$ of the continuum) is added to the synthetic spectrum, with no modification of the state vector.

– Radiometric gain (multiplication): Error in the radiometric accuracy is introduced by apply a $\pm 2\%$ scaling factor to the synthetic spectra.

The magnitude of the instrument errors are defined as the minimum observation requirements for TROPOMI (Landgraf et al., 2016), and again therefore represent the worst case scenarios for instrumentation errors.

These bias effects are investigated for the SWIR1 and SWIR3 bands individually, and are not considered for a combined retrieval. These tests do not cover every possible systematic bias that could be applied (e.g. spectral calibration errors, which can be fitted to reduce errors), however we deem the above tests sufficient to determine the sensitivity of $^{13}CH_4$ retrievals to biases in the a priori information and/or the instrument. Note that the magnitude of the biases applied in this section are identical to those applied in Hu et al. (2016).

Finally we investigate the validity of the light path error cancellation assumption, by comparing the calculated $\delta^{13}C$ values for scattering and non-scattering cases. $\delta^{13}C$ is defined as follows.

$$\delta^{13}C = \left( \frac{\frac{^{13}CH_4}{^{12}CH_4}}{VPDB} - 1 \right) \times 1000\text{\textperthousand}, \tag{2}$$

where VPDB refers to the international standard ratio of $^{13}CH_4$ and $^{12}CH_4$ based on the marine fossil Vienna Pee Dee Belemnite, which has a value of 0.0112372. Sherwood et al. (2016) indicates that biological methane sources such as wetlands typically have $\delta^{13}C$ values of <-60‰, while industrial sources are typically >-30‰.

This comparison will allow us to determine if any bias is present between the calculated values. Errors caused by scattering are not random in nature, but systematic (Inoue et al., 2016). Because this is a synthetic study not based on real data, we can compare the $\delta^{13}C$ values for scattering and non-scattering scenarios using noiseless data. By performing retrievals without instrument noise we remove all random components to calculating the $\delta^{13}C$ ratio, and are left with bias.

## 2.6 Filtering Criteria

Since we are considering a non scattering environment, with no clouds or aerosols there are no filtering criteria applied to the retrievals performed on the synthetic data, in relation to optical depth. All retrievals that fail to converge are filtered, as well as all retrievals which exhibit DFS values lower than unity. All retrievals that show total uncertainties of >3 ppb are also excluded.

## 3 Results - SWIR1

## 3.1 Example spectral fit

First we provide a typical example (shown in Fig. 1) of the spectral fit output from RemoTeC with $^{13}CH_4$ set as the target species. The target species in RemoTeC is retrieved as a profile in 12 pressure equidistant vertical layers, the interfering species ($^{12}CH_4$,$H_2O$, $CO_2$) are retrieved as total column density scalar profiles, assuming a fixed profile shape (Temperature and Pressure are typically not retrieved).

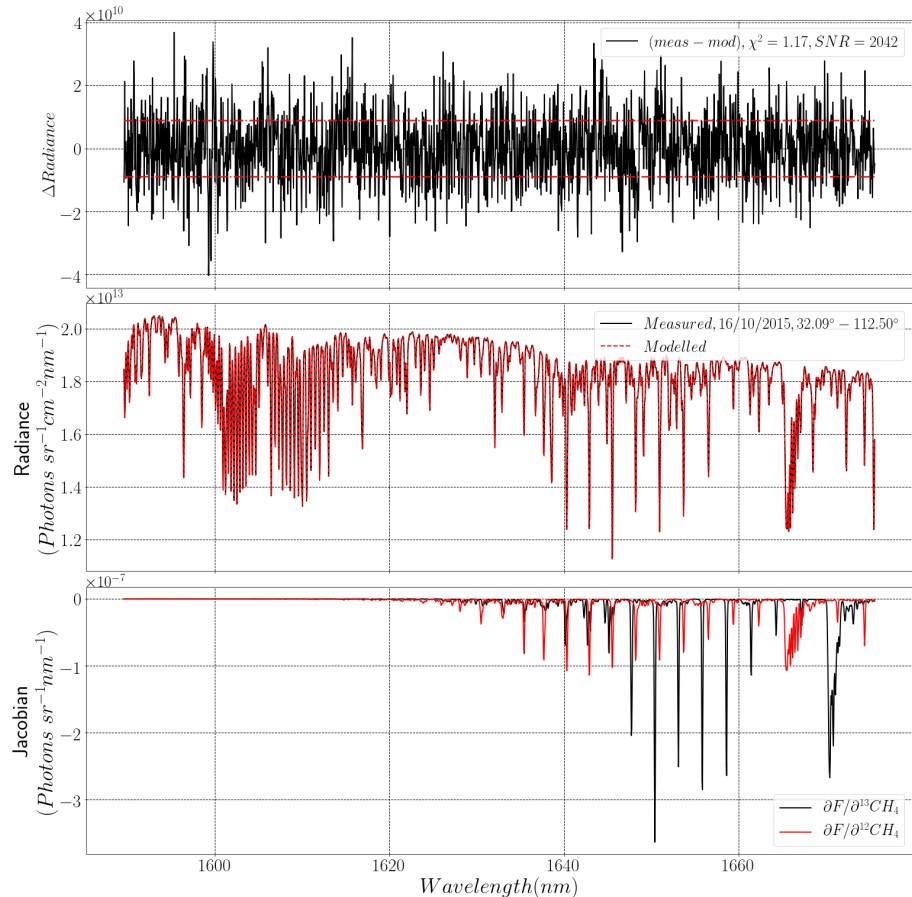

**Figure 1.** Example spectral fit from RemoTeC, assuming the SWIR1 of S5/UVNS. Centre panel: example fit at simulated co-ordinates 32.09°N 112.5°W for a day in October 2015 (black line is synthetic "measured" spectra, red dashed line is retrieved modelled spectra). Top panel shows the spectral residual between modelled and measured, with the red dashed lines indicated the noise level based on the SNR. The bottom panel shows the total column Jacobians of $^{12}CH_4$(red) and $^{13}CH_4$ (black).

The spectral fit quality is good, with a $\chi^2$ value equal to 1.17, and all large spectral residuals are limited to random high frequency components. However, there are some points which could be interpreted as not due to random noise, where the retrieval seems to disagree with the 'truth', notably the methane lines between 1645 and 1650 nm. However, upon further investigation we found that these features do not consistently appear in spectral residuals. Therefore the disagreements shown 5 in Fig. 1 are random in nature.

## 3.2 Averaging kernels

Here we show the column averaging kernel (cAK) for when $^{13}CH_4$ is the target of RemoTeC, in Fig. 2 below. We also show the cAKs for when $^{12}CH_4$ is the target of RemoTeC.

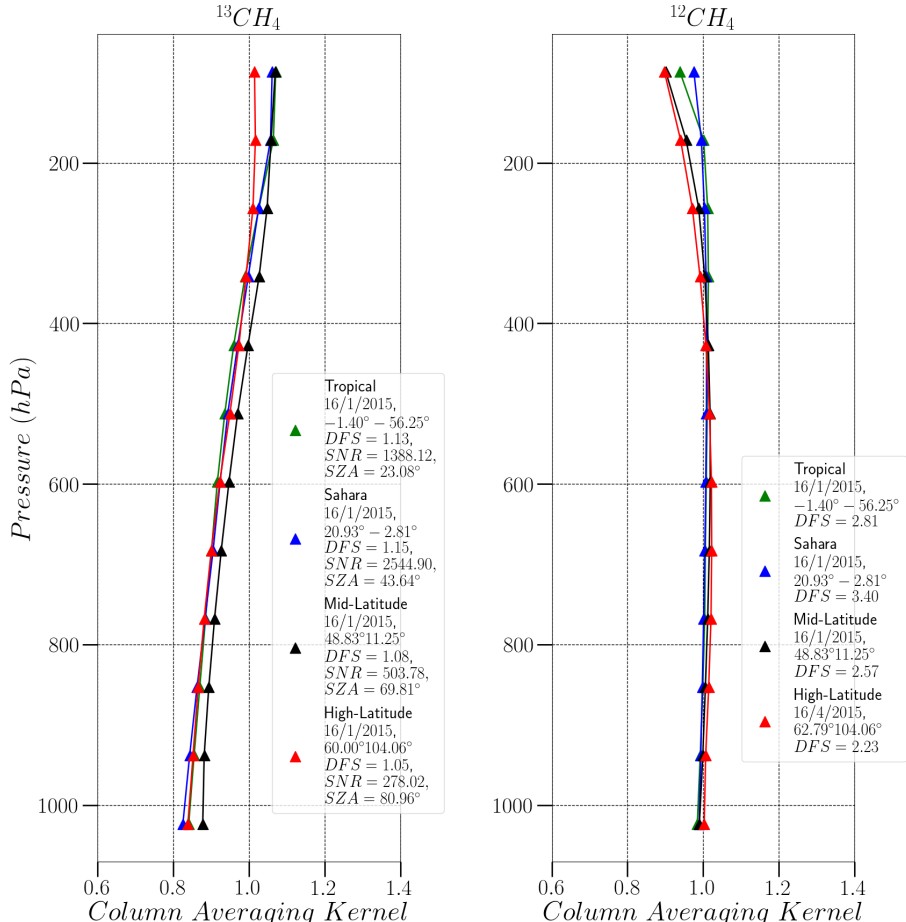

**Figure 2.** Example column averaging kernels from synthetic retrievals of $^{13}CH_4$ (left hand plot) and $^{12}CH_4$ (right hand plot), from the SWIR1 channel of S5/UVNS. The blue plot is an example retrieval over the Sahara desert, the red plot is over Siberia (High-Latitude), the green plot is over the Amazon rainforest (Tropical) and the black plot is temperate Europe (Mid-Latitude). Metadata associated with each retrieval is highlighted in the legend.

Figure 2 shows a tight spread of cAKs which generally do not reach a value of unity in the lower atmosphere, which suggests reduced sensitivity of $^{13}CH_4$ in the lower atmosphere. The cAKs suggest there is significant IC available in total column retrievals of $^{13}CH_4$, but there still may be some noise components present in the retrievals, especially in the lower atmosphere where cAK values are the lowest. The uniformity of the cAKs with respect to surface type, suggest insensitivity to changing atmospheric or retrieval conditions.

The $^{12}CH_4$ cAK exhibits the typical behaviour of $CH_4$ cAKs (e.g. Hu et al. (2016)), which is expected since $^{12}CH_4$ makes up 98% of atmospheric $CH_4$. However the $^{13}CH_4$ cAK exhibits behaviour closer to that of CO (Landgraf et al., 2016). Given that $^{13}CH_4$ makes up $\sim$1.1% of atmospheric $CH_4$, the retrieval column loses sensitivity in the lower atmosphere, where $H_2O$

dominates. Borsdorff et al. (2014) show that in the case where sensitivity is low in the troposphere, the cAK values are enhanced at other altitudes. This is apparent in the cAKs of $^{13}CH_4$ in Fig. 2, where cAK values larger than those of $^{12}CH_4$ are observed.

## 3.3 DFS spread

The next logical step from checking the cAKs is to view the seasonal and geographical distribution of DFS over the synthetic database. This is achieved by plotting the DFS for each retrieval over global maps, as shown in Fig. 3 below. The seasonal dependence will be brought to the fore since we filter out all cases where DFS do not reach unity.

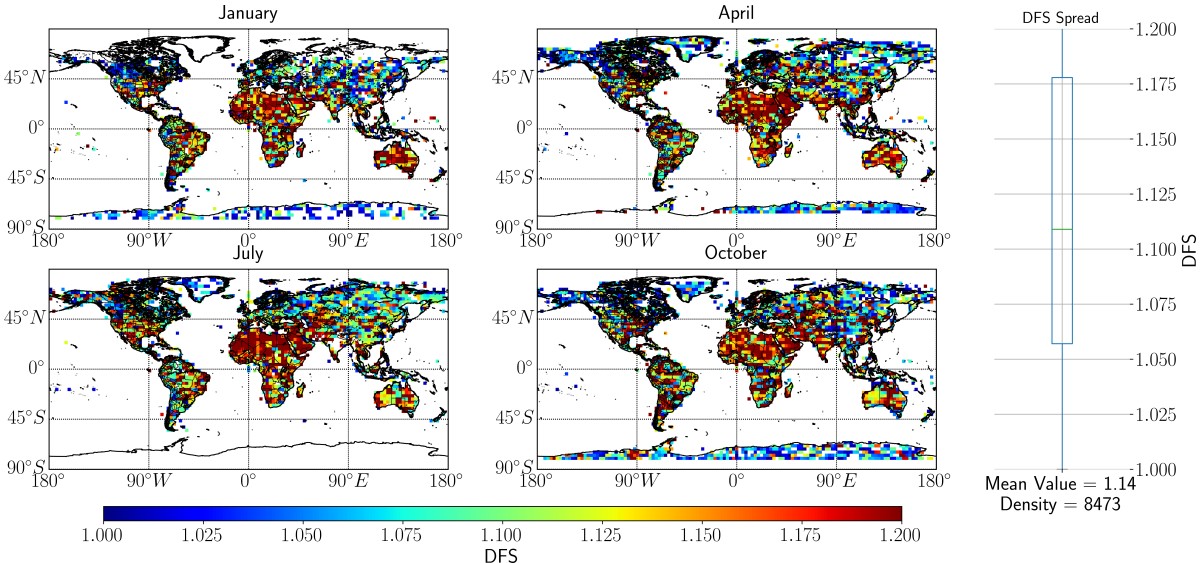

**Figure 3.** Global spread of DFS based on the RemoTeC synthetic data ensemble, with $^{13}CH_4$ as the target for retrievals. The four main seasons in the synthetic database are represented in this figure by one day in the months of January, April, July and October. The far right panel in the figure outlines the spread of DFS values over the entire dataset in the form of a boxplot indicating median and upper and lower quartile values, with the mean value, and the total number of measurements indicated at the bottom, the circles are outlier values.

In Fig. 3 mid-latitude highly reflective surfaces show the highest DFS, and the high latitude/"green" regions showing the lowest DFS values. In general, high information content is achieved with the SWIR1 band, indeed DFS values greater than unity (passing the filtering criteria) are achieved over the Amazon forest regions of Brazil, and for some of the high latitude regions.

## 3.4 Total errors

Section 3.3 suggests that there is enough information in the total column to retrieve $^{13}CH_4$, however this is irrelevant if the retrieval errors are so large as to make assessing $\delta^{13}C$ impossible. The assessed errors from the synthetic database are shown in Fig. 4 below.

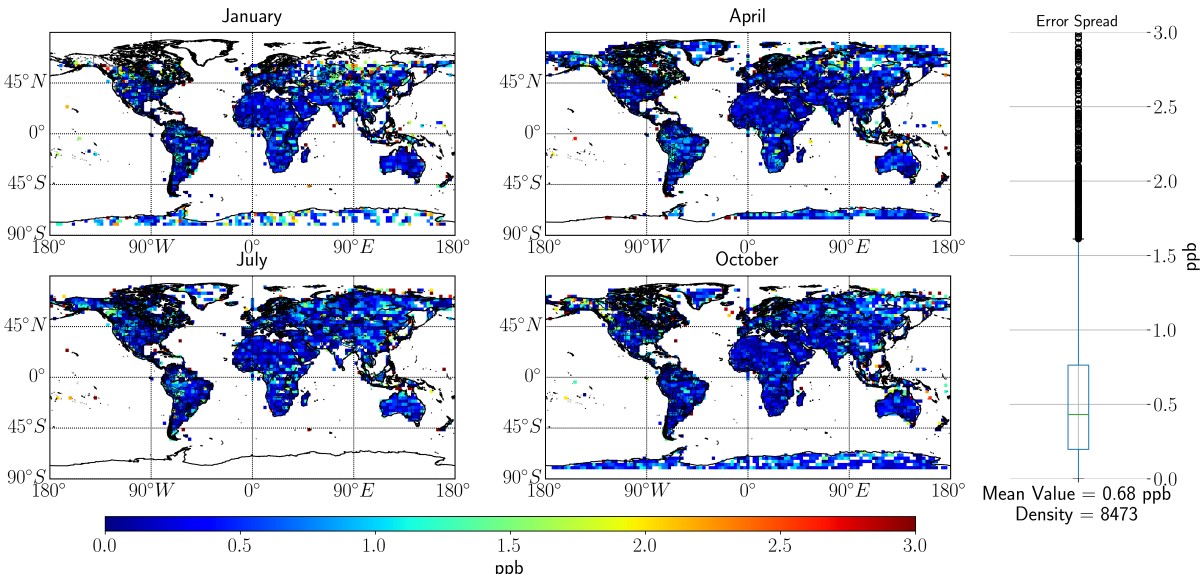

**Figure 4.** Global spread of total error based on the RemoTeC synthetic data ensemble, with $^{13}CH_4$ as the target for retrievals. The four main seasons in the synthetic database are represented in this figure by one day in the months of January, April, July and October. The far right panel in the figure outlines the spread of error values over the entire dataset, with the mean value, and the total number of measurements indicated at the bottom, the circles are outlier values.

Note that the errors in Fig. 4 are remarkably uniform across the seasons and locations (apart from the high latitude regions), suggesting that SNR is not the limiting factor in the SWIR1 band. Typically the mid-latitude errors have values <0.5 ppb, equating to roughly <2.5% of the total column, which we assume to be 20 ppb for $^{13}CH_4$ as indicated in the requirements. The analysis shows that the SWIR1 band results in mean values of 0.68 ppb. Some very high errors in excess of 20 ppb were found

in high latitude regions (when the DFS > 1 and uncertainty < 3 ppb filters were removed), typically within the Arctic circle, but also surprisingly within the south east Asia region. Typically, the largest errors are found in coastal regions where there is likely low albedo causing large errors. An investigation showed that these retrievals are all captured under low SNR conditions, largely driven by SZA and albedo, thus leading to high uncertainty.

Figure A1 shows that the overall majority of the uncertainty can be attributed to precision, with (when considering the mean

uncertainty value) 0.08 ppb uncertainty associated with systematic errors. However this figure makes it clear that the systematic error is larger than the target of 0.02 ppb total uncertainty. Meaning that systematic error must be accounted for before making any judgements on $^{13}CH_4$ retrievals.

### 3.5 Systematic prior knowledge errors

The previous section deals with errors associated with precision, and other systematic errors present in the retrieval approach. In

this section, we investigate the effects of imprecise knowledge of a priori and ancillary information and instrument calibration

errors on the retrieved column of $^{13}CH_4$; for example Fig. 5 below indicates the differences when applying a 2% bias to the a priori methane column.

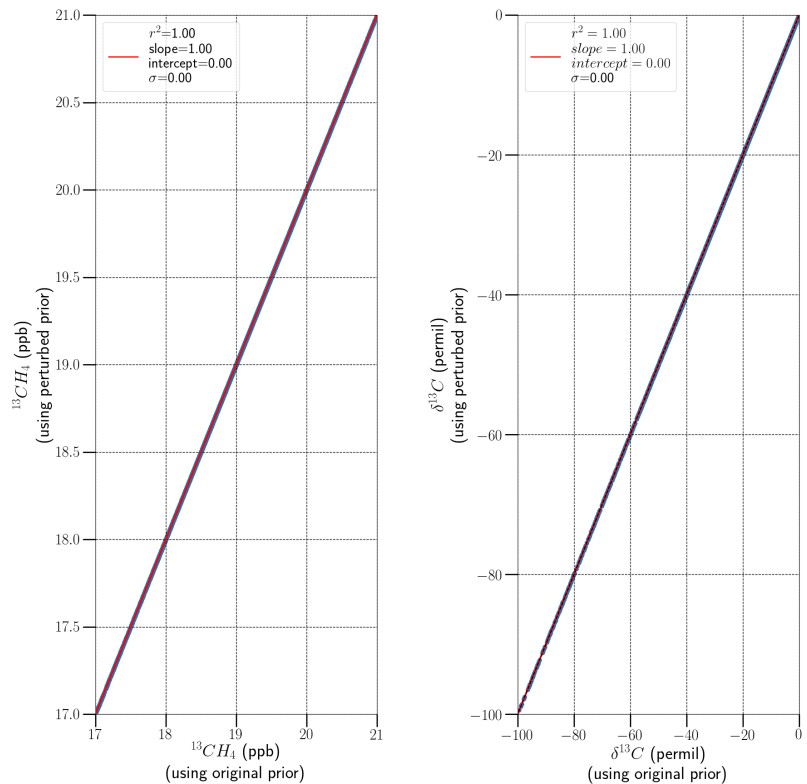

**Figure 5.** Comparison of retrieved $^{13}CH_4$ (left panel) and $\delta^{13}C$ (right panel) before and after a priori modification by applying a 2% bias on the methane a priori. The red line is a line of best fit, and the statistical characteristics (coefficient of correlation, gradient, intercept and standard deviation of the difference) are indicated in the legend.

We show the biases for both $^{13}CH_4$ and $\delta^{13}C$ since the $\delta^{13}C$ ratio is expressed in per mil, and is therefore highly sensitive to any change, in addition to the fact that errors from $^{12}CH_4$ are also included in the $\delta^{13}C$ ratio. In the case of the SWIR1 band we note that a 2% bias in the a priori methane column has no effect on the retrieved a posteriori $^{13}CH_4$ and $\delta^{13}C$ ratio values. Using a similar analysis to that shown in Fig. 5, the bias metrics for the systematic error scenarios described in sect 2.5 are summarised in Table 2, below.

**Table 3.** Effects of errors in a priori databases and instrument calibration errors on test retrievals of SWIR1 $^{13}CH_4$ and $\delta^{13}C$, the metrics displayed in this table are as described in sect 2.5.

| | $R^2$ | $^{13}CH_4$ Slope | Intercept (Bias, ppb) | $\sigma$ (ppb) | $R^2$ | $\delta^{13}C$ Slope | Intercept (Bias, ‰) | $\sigma$ (‰) |
|---|---|---|---|---|---|---|---|---|
| $\Delta CH_4$ = 2% | 1 | 1 | 0.0 | 0.0 | 1 | 1 | 0.0 | 0.00 |
| $\Delta CH_4$ = -2% | 1 | 1 | 0.0 | 0.0 | 1 | 1 | 0.0 | 0.00 |
| $\Delta H_2O$ = 10% | 1 | 1 | 0.0 | 0.0 | 1 | 1 | 0.02 | 0.05 |
| $\Delta H_2O$ = -10% | 1 | 1 | 0.0 | 0.0 | 1 | 1 | -0.02 | 0.04 |
| $\Delta$ T = 2 K | 0.8 | 0.92 | 0.98 | 0.42 | 0.74 | 0.99 | -29.82 | 21.43 |
| $\Delta$ T = -2 K | 0.8 | 1.05 | -0.27 | 0.47 | 0.67 | 0.95 | 33.43 | 24.06 |
| $\Delta$ P = 0.3% | 1 | 1 | -0.02 | 0.01 | 1 | 1 | 0.65 | 0.46 |
| $\Delta$ P = -0.3% | 1 | 1 | 0.03 | 0.01 | 1 | 1 | -0.64 | 0.47 |
| Offset = 0.1% | 1 | 1 | 0.01 | 0.02 | 1 | 1 | 4.41 | 1.21 |
| Offset = -0.1% | 1 | 1 | -0.01 | 0.02 | 1 | 1 | -4.41 | 1.21 |
| Gain = 2% | 1 | 1 | 0.0 | 0.0 | 1 | 1 | 0.0 | 0.03 |
| Gain = -2% | 1 | 1 | 0.0 | 0.0 | 1 | 1 | 0.0 | 0.02 |

The systematic errors indicated in Table 3 suggest that uncertainty in the a priori state vector do not adversely affect $\delta^{13}C$ calculations, however uncertainty in the pressure and temperature ancillary data do have a notable impact, which translates to large biases in $\delta^{13}C$ values. This impact could be reduced when averaging over monthly periods, since pressure and temperature errors are unlikely to be systematically offset over a long period. We therefore assume that pressure errors are of lesser relevance. However, the 2 K temperature error still results in a bias of roughly -30‰, and scatter of roughly 26‰. This amount of bias renders the usefulness of retrieving the $\delta^{13}C$ ratio considerably. Reuter et al. (2012) describe how the lower state energy ($E_0$) of molecular transitions fundamentally controls the temperature sensitivity for each molecule in relation to carbon dioxide isotopologues. The HITRAN2012 database shows that the $E_0$ values for $^{13}CH_4$ are typically several times lower than the main methane isotopologue, and therefore will be affected by a temperature shift to a greater degree than the main methane isotopologue. The exponential relationship between the lower state energy and the line strength (Eq. 3) suggests that molecules with lower $E_0$ values (such as $^{13}CH_4$) are much more affected by temperature shifts in the cross sections, as opposed to molecules with higher $E_0$ values such as $^{12}CH_4$. An et al. (2011) show that for a given temperature difference, the change in line intensity can be expressed as

$$\frac{S(T)}{S(T_0)} = \frac{Q(T_0)}{Q(T)}exp(-\frac{hcE_0}{k}(\frac{1}{T}-\frac{1}{T_0})), \tag{3}$$

where $S(T)$ is the line intensity, $Q(T)$ is the total partition function of the absorbing molecule and T is temperature. Note that RemoTeC includes the option for fitting a temperature offset, the results of including this option in the retrieval process are shown in Fig. 6 below.

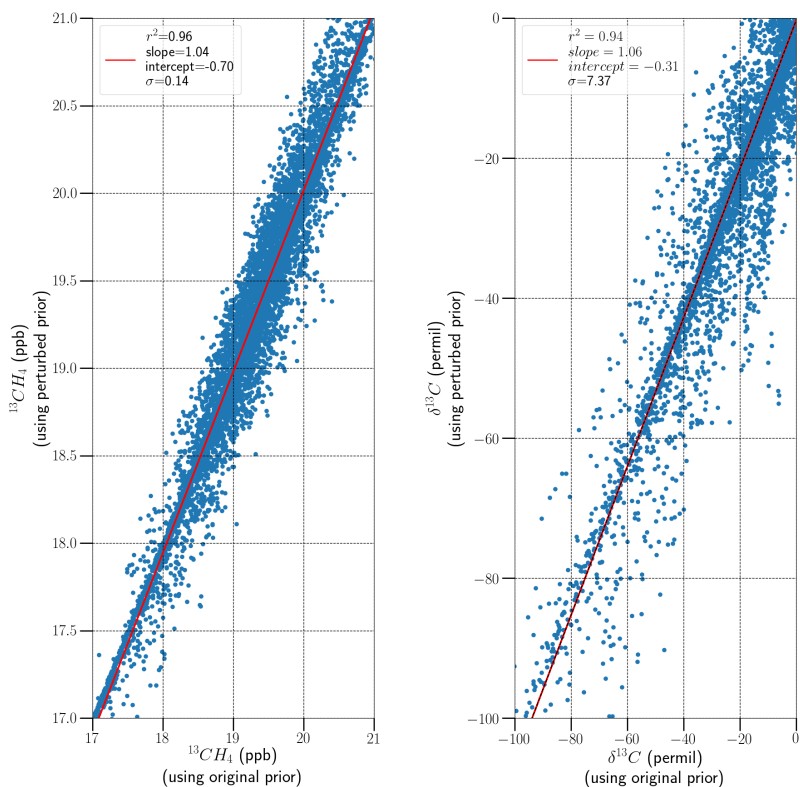

**Figure 6.** Comparison of retrieved $^{13}CH_4$ (left panel) and $\delta^{13}C$ (right panel) before and after a priori modification by applying a 2 K bias on the temperature a priori. Where temperature shift is retrieved as a part of the state vector. The red line is a line of best fit, and the statistical characteristics (coefficient of correlation, gradient, intercept and standard deviation of the difference) are indicated in the legend.

Figure 6 shows that including the temperature shift reduces the temperature sensitivity bias of the $\delta^{13}C$ retrievals to -0.31‰
5 but comes at the cost of reducing valid retrievals by 50%, which fits within the set requirements of this study. However, note that the scatter on the temperature fitted retrieval is significant (>7‰), suggesting that temporal and or spatial averaging is required. The preferable solution here is to improve knowledge of the ancillary information, rather than rely on longer term spatio-temporal averaging.

In addition to the errors in the a priori and ancillary profiles, we note that the apparent sensitivity to radiometric offset errors, where $\pm 0.1\%$ causes a $\delta^{13}C$ bias of up to 4.41‰, is highly significant. This is likely an effect of the high SNR achievable in the SWIR1 band.

## 3.6 Summary of SWIR1

The results shown for the planned SWIR1 band in UVNS indicate a difficult but positive outlook for the future. The SWIR1 band shows global DFS and errors are uniform across the globe, aside from high latitude regions, with high reflectance regions such as the Sahara desert showing similar patterns in DFS and total errors, as lower reflectance regions such as the Amazon in south America. For a mean error of 0.68 ppb (or 0.57 ppb for precision), a daily repeat cycle could theoretically lead to the desired precision of 0.02 ppb in 2-3 years assuming instantaneous measurements. This is not useful, but given the planned high spatial resolution of S5/UVNS, it would be easy to expand the spatial averaging in order hit the target precision within one year of averaging. Further, some of the mid-latitude regions already have errors that have low precision errors <0.2 ppb, and theoretically the target of 0.02 ppb errors could be achieved with one month of averaging with a larger spatial sample, which would be helpful to monitor the annual change of $\delta^{13}C$ within a specific region.

However, the sensitivity of $^{13}CH_4$ retrievals to temperature, and instrument errors will likely mean that total $\delta^{13}C$ uncertainty is significantly higher, and assessments on the accuracy of the temperature and pressure will likely be required to make judgements on the required level of spatial and/or temporal averaging required. The instrument sensitivity can be assessed after launch, and removed from the spectra prior to full retrievals, and thus remove $^{13}CH_4$ sensitivity to instrument errors (accepting that this will be challenging). Generally the SWIR1 band looks to be suited to $^{13}CH_4$ retrieval, with some potential to track $\delta^{13}C$ over the course of a year once systematic and a priori bias corrections are applied.

## 4 Results - SWIR3

### 4.1 Example spectral fit

Following the same format as that shown in sect. 3., an example of a spectral fit in the SWIR3 band is shown in Fig. 7 below.

The quality of the fit shown in Fig. 7 is similar to that shown in Fig. 1, with the residual radiance showing similar values based on the $\chi^2$ value. However, it is important to note that the spectral lines for both the simulated spectra and retrievals are based on Voigt line shapes, which, although resulting in good fits in simulated scenarios, may not be adequate in reality and could cause worse fits. This also applies to other errors that may be present in the fitting process (e.g. ILSF or similar). Note that the radiance magnitudes in this spectral region are significantly lower than the equivalent radiances in Fig. 1, which is not unexpected since solar irradiance and surface albedo in this waveband is significantly lower than in SWIR1. This is best indicated by the SNR shown in the top panel of Fig. 7, which is several times smaller than the equivalent in Fig. 1. Reuter et al. (2010) note that measurements of $CO_2$ in the SWIR1 spectral region with SCIAMACHY tend to have SNR values between 279 and 1950.

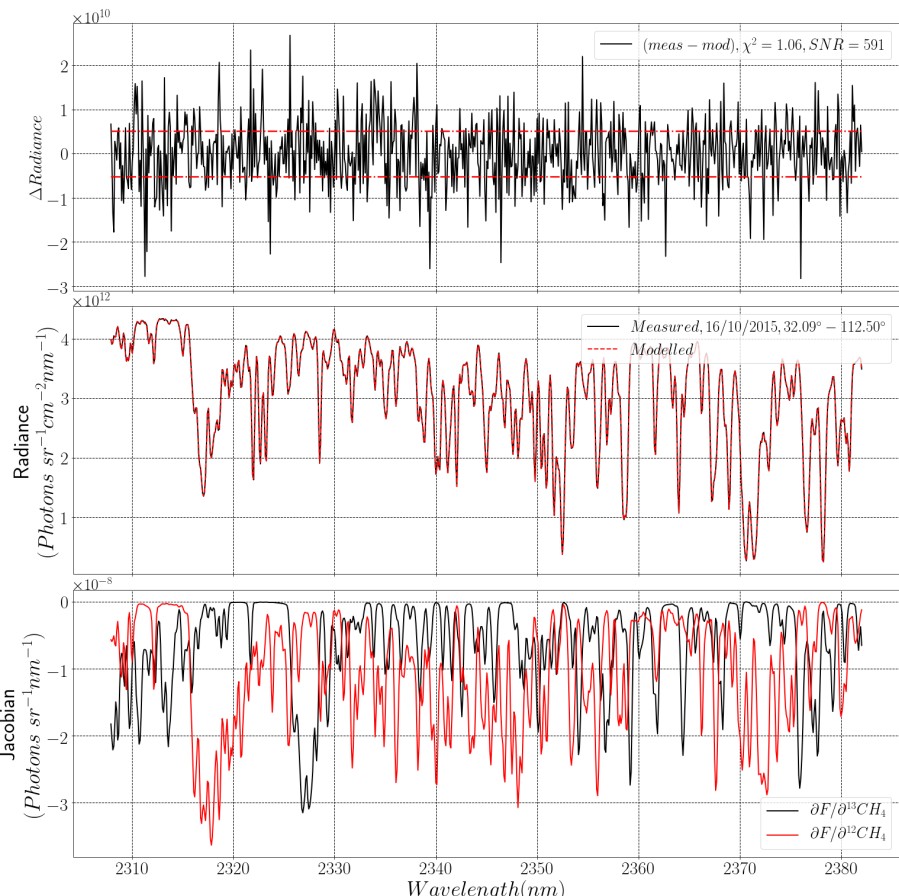

**Figure 7.** As Fig. 1, but focused on the TROPOMI/UVNS SWIR3 band.

The Jacobians in the bottom panel suggest more $^{13}CH_4$ spectral lines in this waveband as compared to SWIR3 (Fig. 1 above). However, the Jacobians in Fig. 7 appear to be more dominated by $^{12}CH_4$, since this spectral range is closer to a methane continuum than to a collection of individual spectral lines, as is found in SWIR1.

## 4.2 Averaging kernels

5   Hu et al. (2016) show an example of a total column averaging kernel (cAK) for $CH_4$ retrievals from TROPOMI, with the values remaining close to unity for the total column, thus implying that the TROPOMI SWIR methane retrievals maintain high sensitivity throughout the total column. Here we show the equivalent column averaging kernel for when $^{13}CH_4$ is the target of RemoTeC, in Fig. 2 below. We also show the equivalent cAKs for when $^{12}CH_4$ is the target of RemoTeC. $^{12}CH_4$ cAKs should show similar behaviour to the cAKs of Hu et al. (2016).

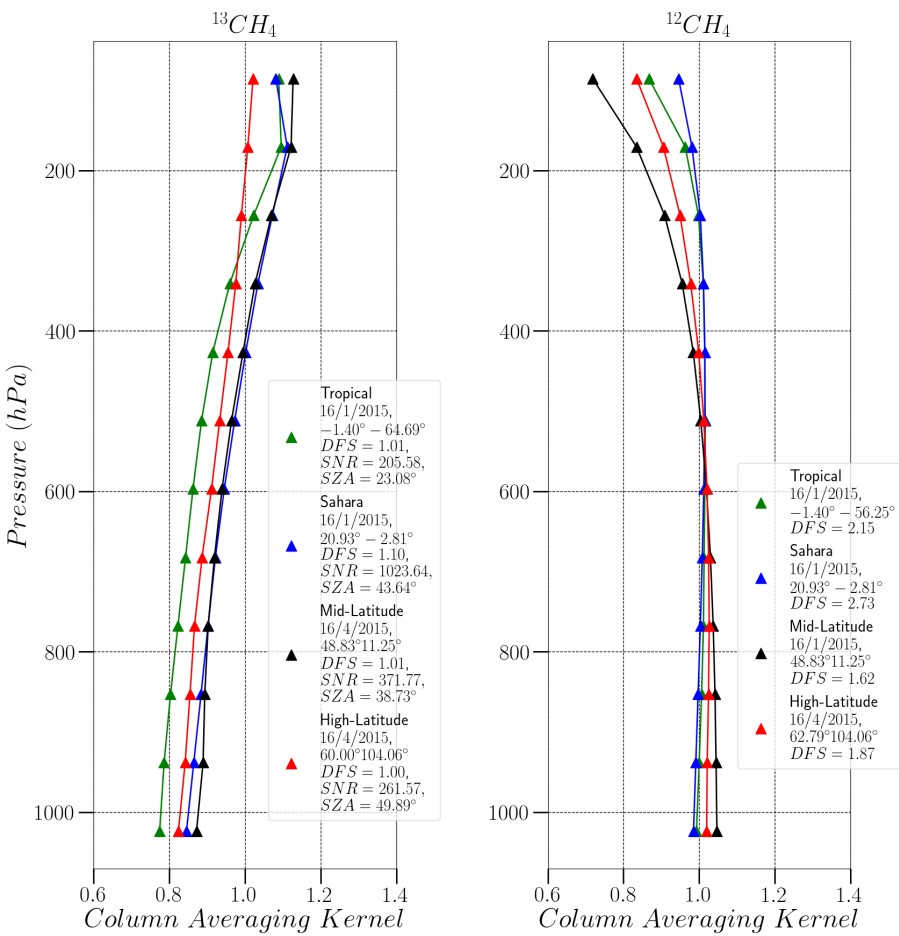

**Figure 8.** As Fig. 2, but focused on the Sentinel 5/5P SWIR3 band.

The cAKs in Fig. 8 shows very similar total column shapes to the SWIR1 cAKs, i.e. weaker in the lower at atmosphere, and stronger in the upper atmosphere. The shape of the cAKs is almost the mirror image of the example cAK shown by Hu et al. (2016), however the cAKs of CO retrieval shown by Landgraf et al. (2016) show similar shape cAKs, suggesting that weak atmospheric absorbers struggle for information content in the lower atmosphere (where spectroscopy effects such as pressure broadening likely make it difficult for weak absorbers). In addition the SWIR1 cAKs typically have higher magnitudes, suggesting higher information content in the retrievals and less impact by pressure broadening and similar effects.

### 4.3 DFS spread

The global spread of DFS values for the SWIR3 band are shown in Fig. 9 below.

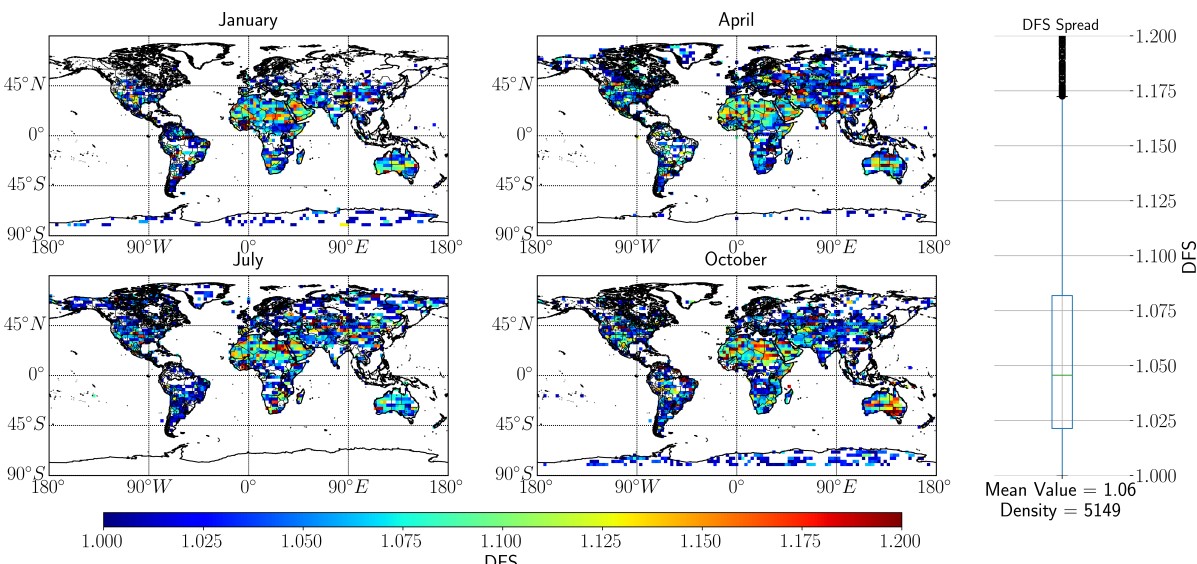

**Figure 9.** As Fig. 3, but focused on the Sentinel 5 SWIR3 band.

Figure 9 suggests that DFS values of unity or better can be expected for mid-latitude regions in all seasonal conditions, however high latitude regions such as Antarctica or Greenland may not achieve DFS values of unity, in Winter and Autumn most likely due to a combination of low surface reflectance, and high solar zenith angles (SZA). In addition, we see that the highest DFS values typically occur in desert regions such as the Sahara or Arabian peninsula, and the Amazon rain forest tends

5 not to achieve unity values at most times of year.

The differences in measurement densities indicated in both Figs 3 and 9, show that the SWIR1 band has almost 3000 additional valid retrievals that pass the filtering criteria as opposed to SWIR3. These additional valid retrievals represent roughly 30% of the total ensemble, and is therefore a significant proportion.

## 4.4   Total errors

10   The global spread of total retrieval errors is shown in Fig. 10 below.

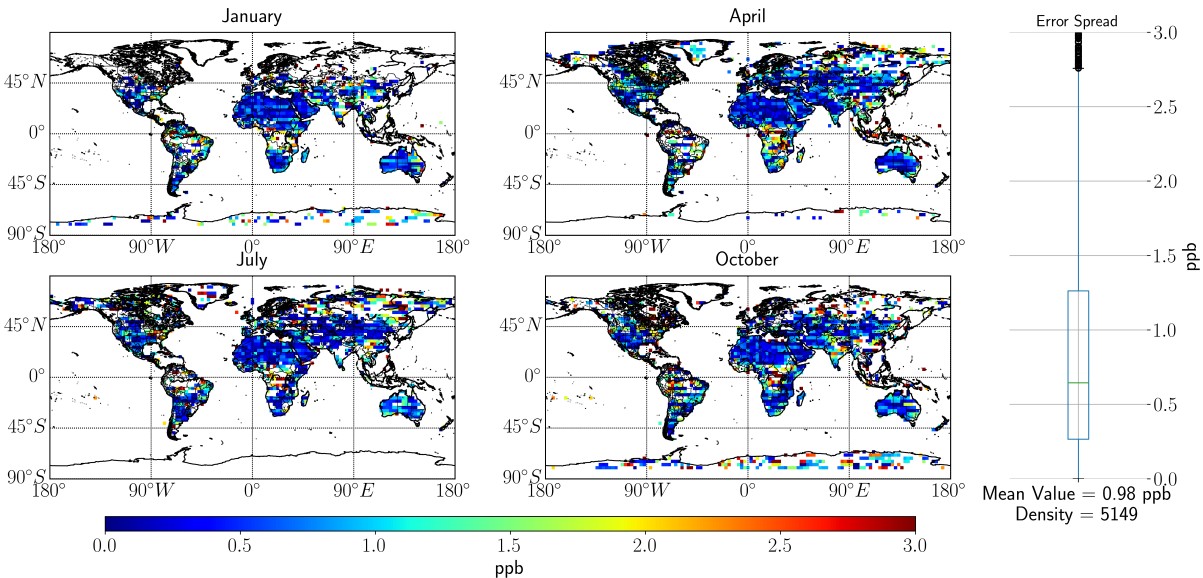

**Figure 10.** As Fig. 4, but focused on the Sentinel 5/5P SWIR3 band.

As expected (for non scattering scenarios), Fig. 10 shows that the minimum errors occur in the high DFS regions shown in Fig. 9. These regions show that total errors typically range between 0.5 and 1.0 ppb (although some cases where errors > 3 ppb, normally in tropical/sub-tropical regions when filters were removed), which equates to roughly between 2.5-5% total column error which is remarkable for such a minor species. The plot of the spread of errors suggests a mean value of ~1 ppb over the entire year, considering all surface types. When we removed the unity DFS filtering criterion, our investigation found some regions had errors exceeding 20 ppb, typically in high latitude/low albedo regions such as Greenland.

The precision error map shown in Fig. A2 indicates similar levels of systematic error in the SWIR3 when compared to the SWIR1 band.

## 4.5 Systematic prior knowledge errors

Following the methods laid out in sects. 2.5 and 3.5, the following section investigates the effects of uncertainty in the prior state vector and ancillary information on $^{13}CH_4$ retrievals in the SWIR3 band. Like in sect. 3.5 we show the biases for both $^{13}CH_4$ and $\delta^{13}C$, due to their differences in sensitivity to the perturbations. These are highlighted in Table 4, below.

**Table 4.** Effects of errors in a priori databases and instrument calibration errors on test retrievals from SWIR3 of $^{13}CH_4$ and $\delta^{13}C$, the metrics displayed in this table are as described in sect 2.5.

| | | $^{13}CH_4$ | | | | $\delta^{13}C$ | | |
|---|---|---|---|---|---|---|---|---|
| | $R^2$ | Slope | Intercept (Bias, ppb) | $\sigma$ (ppb) | $R^2$ | Slope | Intercept (Bias, ‰) | $\sigma$ (‰) |
| $\Delta CH_4$ = 2% | 1 | 1 | 0.01 | 0.02 | 1 | 1 | 0.81 | 1.02 |
| $\Delta CH_4$ = -2% | 1 | 1 | 0.01 | 0.02 | 1 | 1 | 0.81 | 1.02 |
| $\Delta H_2O$ = 10% | 1 | 1 | 0.01 | 0.02 | 1 | 1 | 0.75 | 0.96 |
| $\Delta H_2O$ = -10% | 1 | 1 | 0.01 | 0.02 | 1 | 1 | 0.87 | 1.09 |
| $\Delta$ T = 2 K | 0.72 | 1.06 | 0.39 | 0.73 | 0.58 | 0.89 | 73.82 | 36.88 |
| $\Delta$ T = -2 K | 0.74 | 0.86 | 1.42 | 0.55 | 0.72 | 1 | -68.33 | 27.79 |
| $\Delta$ P = 0.3% | 1 | 0.99 | -0.03 | 0.04 | 1 | 1 | -4.74 | 2.20 |
| $\Delta$ P = -0.3% | 1 | 1.01 | 0.06 | 0.07 | 1 | 0.99 | 6.30 | 3.64 |
| Offset = 0.1% | 1 | 1 | 0.01 | 0.02 | 1 | 1 | 0.81 | 1.02 |
| Offset = -0.1% | 1 | 1 | 0.01 | 0.02 | 1 | 1 | 0.81 | 1.02 |
| Gain = 2% | 1 | 1 | 0.01 | 0.02 | 1 | 1 | 0.81 | 1.02 |
| Gain = -2% | 1 | 1 | 0.01 | 0.02 | 1 | 1 | 0.81 | 1.02 |

The results in Table 4 typically show that including systematic biases for most of the considered parameters have similar magnitudes, with two notable exceptions, pressure and temperature. The 0.3% pressure bias induces up to roughly 6‰ bias in the $\delta^{13}C$ values, thus making the 10‰ target more challenging. However the main issue is the sensitivity to temperature, with a 2 K error resulting in biases of over 60‰ $\delta^{13}C$. The reasons for this temperature sensitivity likely stem from Eq. (3), shown in sect 3.5 above. The temperature bias effects can be reduced if a temperature shift is included in the state vector, the results of which are shown in Fig. 11.

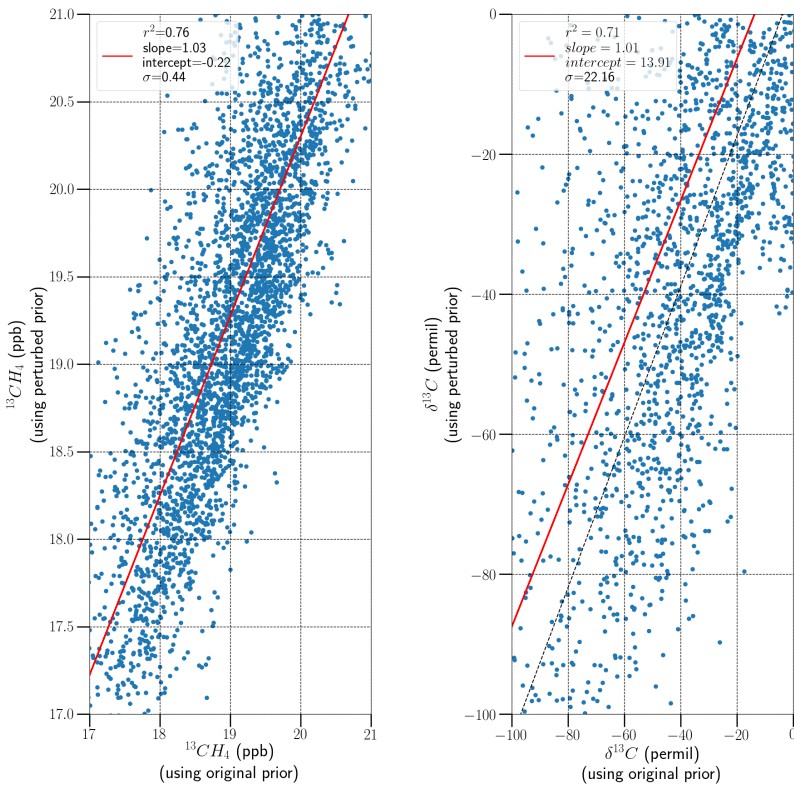

**Figure 11.** As Fig. 5, but including a 2 K temperature bias in the retrieval process, and setting RemoTeC to account for temperature offsets.

Comparing the results in Fig. 11 with those shown in Table 4 show a significant improvement in the bias, but this has come at a cost of the quality of the fits where we found that ~50% of the synthetic scenarios failed to converge, as compared to 99% convergence before enabling the temperature fitting. However, even with this improvement in bias, the magnitude of the bias (13.9‰) is still greater than the desired magnitude of the total error on the $\delta^{13}$C metric. Again like in the SWIR1 band, the

5    preferable solution would be to have more accurate knowledge of temperature. Therefore at this time, the current RemoTeC algorithm needs more accurate knowledge of temperature (and pressure) profiles before meaningful values of $\delta^{13}$C can be generated in this spectral band. It could be argued that it may be possible to average out this temperature bias over time, since it is unlikely that the temperature profile will be systematically offset as much as $\pm$2 K over a period of time, but this is very difficult to predict and is not a solution to rely upon. An additonal difficulty with this temperature dependance, as Reuter et al.

10    (2012) identify, is that such temperature sensitivity can add significant uncertainty to the light path at which point the light path proxy method becomes a much less effective method for removing scattering effects.

Radiometric offset errors are less significant in the SWIR3 as opposed to the SWIR1 band. Radiometric offset errors typically lead to underestimation in the surface reflectance estimate in the absence of aerosols/clouds (Kuze et al., 2014). Given that the SWIR1 operates at a higher radiance magnitude than the SWIR3, any minor errors in the surface reflectance estimate will likely lead to larger errors in the calculated radiance, as opposed to the true radiance. Thus leading to larger errors in the SWIR1.

## 4.6    Summary of SWIR3

We find that in principle, retrieval of $^{13}CH_4$ using the SWIR3 band of TROPOMI/UVNS is feasible, with all regions of the globe showing DFS in the region of unity, exemplified by uniformity in the cAKs, which show typical responses for weak absorbers. Errors vary significantly, but are typically at their minimum over desert or high reflectance scenes and maximum over "green" scenes or high latitude regions, indicating that the quality of the retrievals is heavily dependent on SNR. Individual retrieval errors are too high to hit the basic error target of 0.02 ppb, however, as with SWIR1 the precision error can be reduced through temporal averaging. The low error regions (typically <0.5 ppb) can achieve a precision of 0.02 ppb with roughly 1-2 years of instantanious measurements (assuming none are corrupted by clouds or similar). However, if we consider mean error values of 1 ppb, the target precision of 0.02 ppb becomes harder to achieve, with multi-year datasets required, even if spatial averaging is taken into account. Note that these values are very optimistic, since they do not take into account errors in the retrieval of $^{12}CH_4$, or the fact that retrievals may fail due to the presence of aerosols. Hu et al. (2016) estimate that approximately 50% of the synthetic measurements are not valid due to high aerosol optical depth, which means that we effectively have to double our temporal averaging period. However, all of these points are moot when considering the high systematic error caused by poor knowledge in the temperature profiles. Therefore, while there is enough information content in the total column retrievals of $^{13}CH_4$, and the precision errors could be low enough to make calculating $\delta^{13}C$ a worthwhile task, very accurate prior knowledge of the state vector and ancillary elements is required. When comparing with the SWIR1, we find a reduced performance thus implying SNR is a more important factor in $^{13}CH_4$ retrieval than spectral resolution (when comparing with Malina et al. (2018), who investigated methane isotopologue retrieval assuming GOSAT-2/TANSO-FTS-2 characteristics of spectral resolution 0.2 cm$^{-1}$). The key reasons for the lower SNR in the SWIR3 band as opposed to SWIR1 are the lower solar irradiance and surface albedo at these wavelengths.

# 5    Results - SWIR1 + SWIR3

Finally, we consider the potential benefit of a dual band retrieval of $^{13}CH_4$, assuming normal operations of UVNS and not a specialised mode of operation. This section shows results from a dual band retrieval in a purely algorithmic sense.

## 5.1    Averaging kernels

The cAKs for the combined SWIR1 and SWIR3 bands are shown in Fig. 12 below.

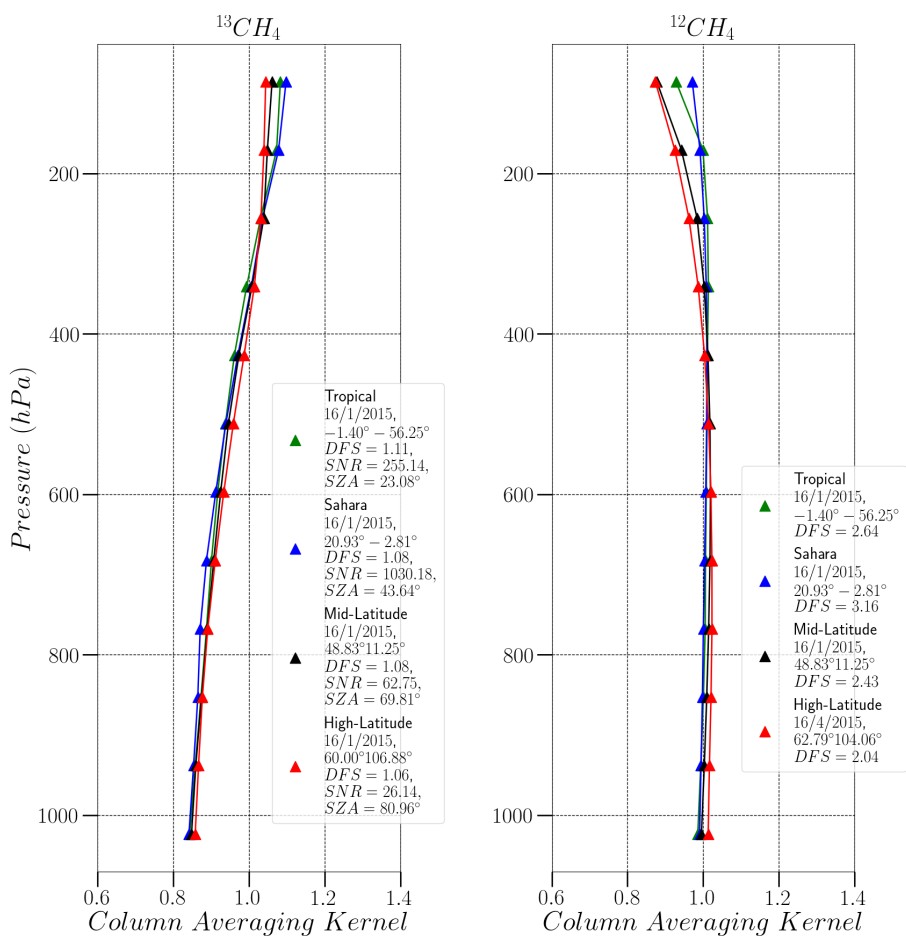

**Figure 12.** As Figs. 2 and 8, but focused on a combination of the SWIR channels from Sentinel 5.

The results in Fig. 12 show characteristics most closely aligned with the SWIR1 band considered on its own. In general however, the cAKs shown in Figs 2, 8 and 12 are all similar, and there are only minor variations between the example retrievals and bands.

## 5.2 DFS spread

5  The global spread of DFS values generated by combining the SWIR1 and SWIR3 channels is shown in Fig. 13 below.

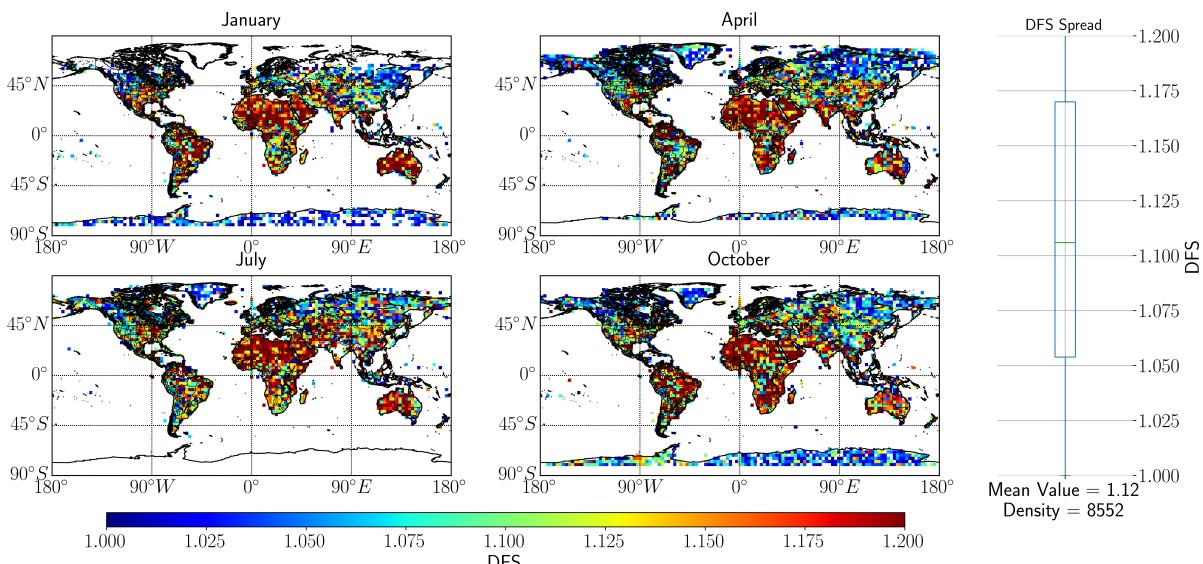

**Figure 13.** As Figs. 3 and 9, but focused on a combination of the SWIR channels from Sentinel 5.

Spread and magnitude are very similar to those for SWIR1 alone (Fig. 3), again suggesting that the information content from the SWIR1 channel dominates the dual band retrieval. Note that the mean DFS value for the dual band method is lower than that for the SWIR1 band alone, however, the dual band method includes additional valid retrievals in the higher latitude regions, which likely account for the lower mean DFS.

5 **5.3 Total errors**

The total errors for the dual band retrieval are shown in Fig. 14 below.

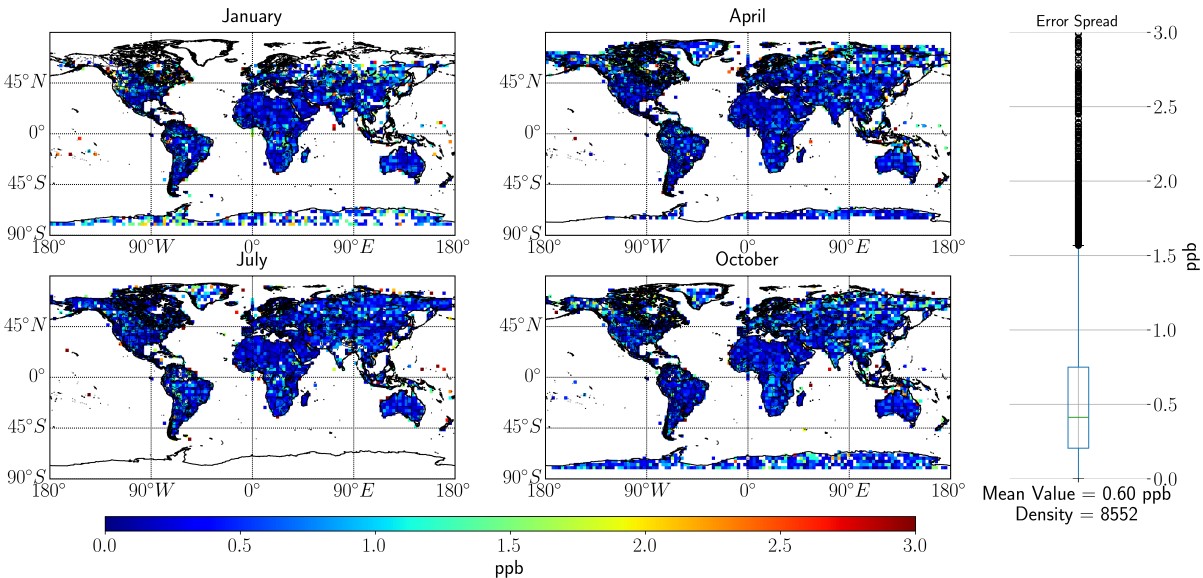

**Figure 14.** As Figs. 4 and 10, but focused on the Sentinel 5 SWIR1 and SWIR3 bands.

The total errors show a minor improvement in the total column $^{13}CH_4$ errors, with a mean value 0.08 ppb lower than for the SWIR1 band on its own. Similarly to Fig. 13, this minor improvement is caused by higher IC, but tempered by the additional valid retrievals in the high latitudes, not present in the SWIR1 band, and which typically have larger errors. In general, these differences are minor, and it is clear that the dual band retrieval has only a small effect on the retrieval errors. The benefits

are very likely to vanish when considering the combination of the systematic errors indicated in Tables 3 and 4, and other instrument or physical errors associated with a dual band retrieval.

The dual band retrieval precision errors shown in Fig. A3 indicate that systematic error magnitudes are similar to those for each band considered separately.

## 6 Scattering vs non-Scattering Comparison

When making comparisons between scattering and non scattering cases, we found that scattering has a limited impact on the synthetic retrievals. We state in sect. 2.3 that treatment of aerosols in the synthetic scenarios is more advanced than the retrieval algorithm. The deliberate difference in the level of sophistication of the aerosol representation between the synthetic spectra and the retrieval algorithm means that additional forward model errors will be introduced into the retrievals. Below we show how the retrieved AOD relates to the retrieval errors for SWIR1. Blank areas in the figures below are areas where the retrievals

have failed to converge.

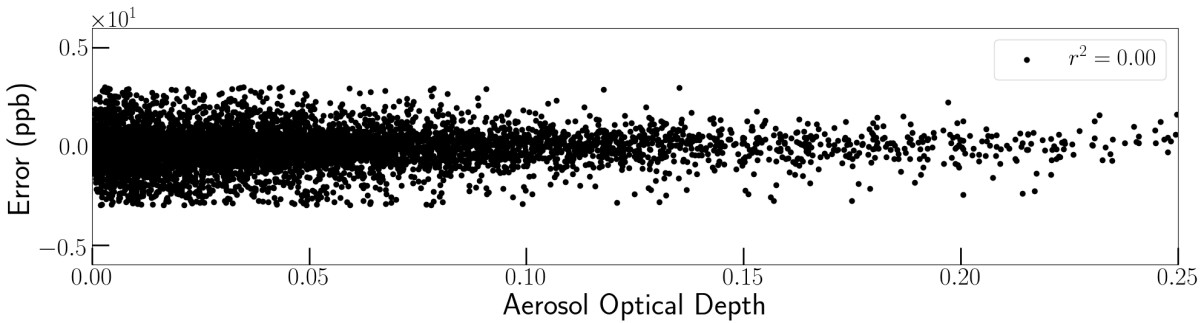

**Figure 15.** Scatter plot of retrieved AOD vs total uncertainty in the $^{13}CH_4$ column from the SWIR1 band. The coefficient of determination is indicated in the legend.

Figure 15 indicates that there is no relationship between uncertainty and AOD. We explore this further in the following subsections by comparing $\delta^{13}C$ calculated with no aerosols and no scattering, with $\delta^{13}C$ calculated including scattering with aerosols. If biases due to scattering are present, we would expect to see large differences in high aerosol regions (e.g. the Sahara).

## 6.1 SWIR1

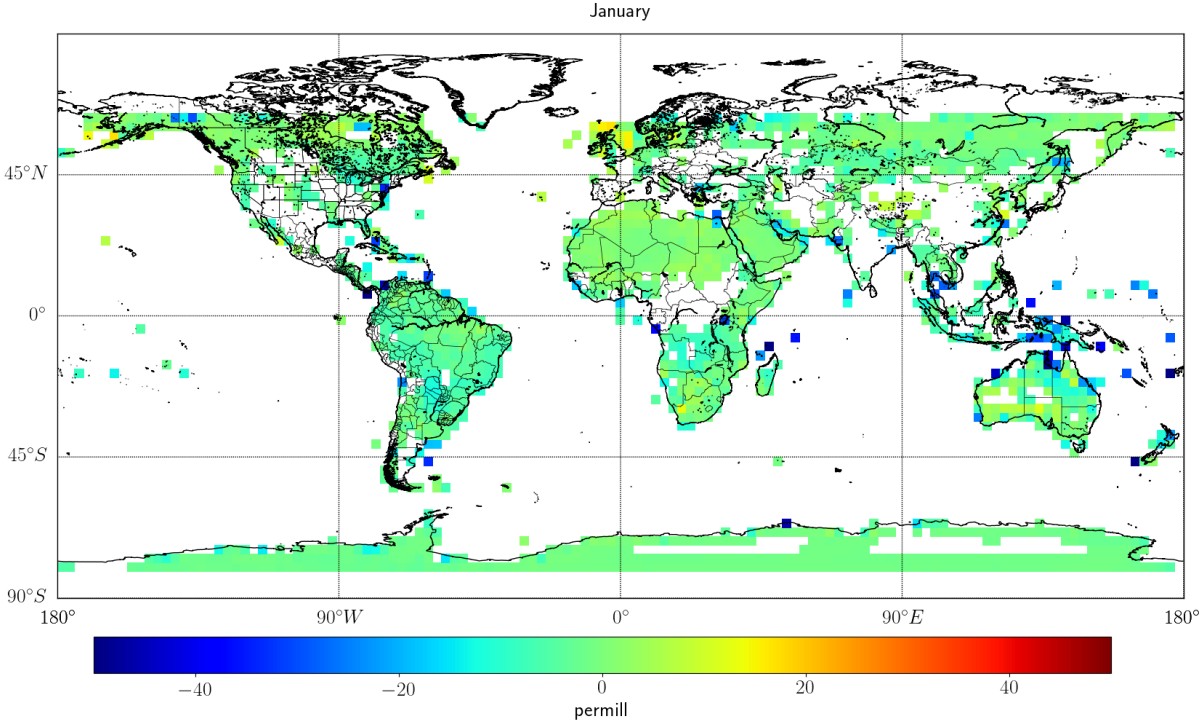

**Figure 16.** Difference in $\delta^{13}$C values from non scattering retrievals and scattering retrievals without instrument noise. Figure represents a day in the month of January 2015 for SWIR1 retrievals.

Figure 16 above shows that there is no significant global bias between the scattering and non-scattering cases. The bias appears to be small (roughly a global mean value of roughly -4‰, which is small in comparison to the uncertainties shown in the sections above, but also largely uniform across the globe with some notable exceptions (e.g. high latitude regions, Hudson's bay in Canada and in the Pacific ocean). These regions are not known for having significant aerosol errors. This suggests that the apparent systematic bias is not due to the presence of aerosols or caused by scattering.

## 6.2 SWIR3

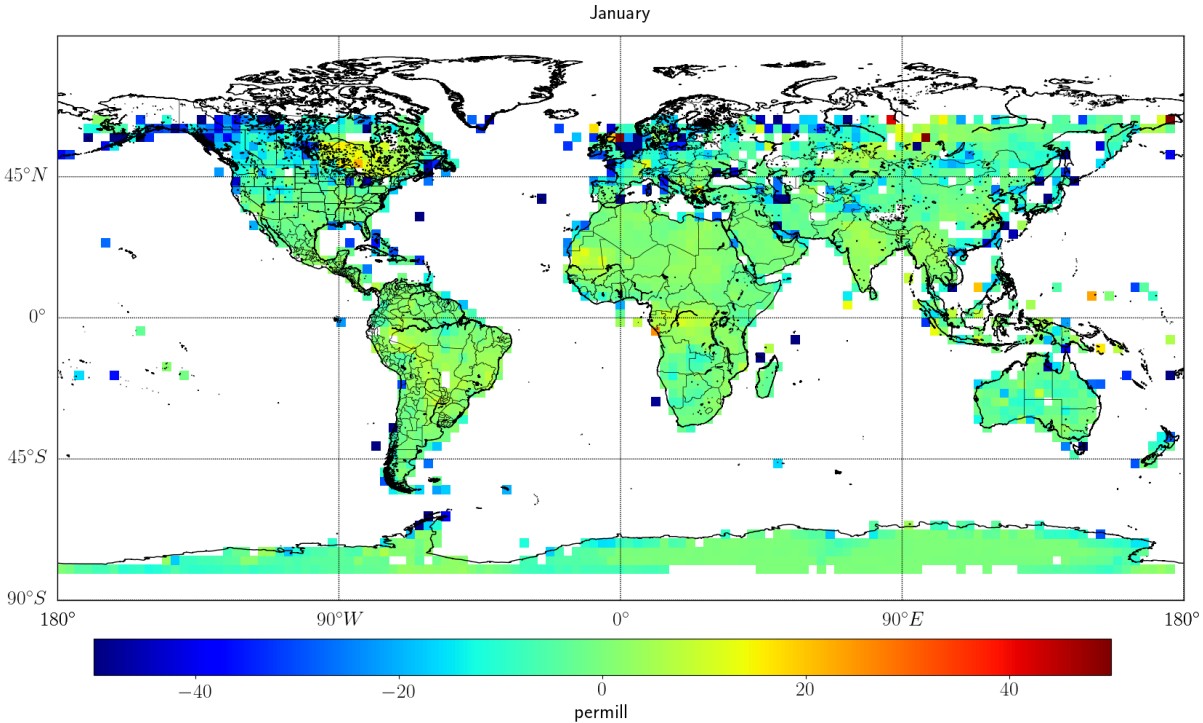

**Figure 17.** As Fig. 16, but focused on the common SWIR3 band.

We note that the $\delta^{13}$C ratio differences represented in Fig. 17 are very similar to those shown in Fig. 16, including the pronounced spike in systematic error over Hudson's bay in Canada, and in the Pacific. The mean difference is roughly -4‰, again similar to Fig. 16. The SWIR3 band is known to be less affected by scattering than the SWIR1 band, and should therefore exhibit lower systematic bias than Fig. 16, if the differences are caused by scattering. This suggests that scattering is not responsible for the biases present in Fig. 17.

## 6.3 SWIR1 + SWIR3

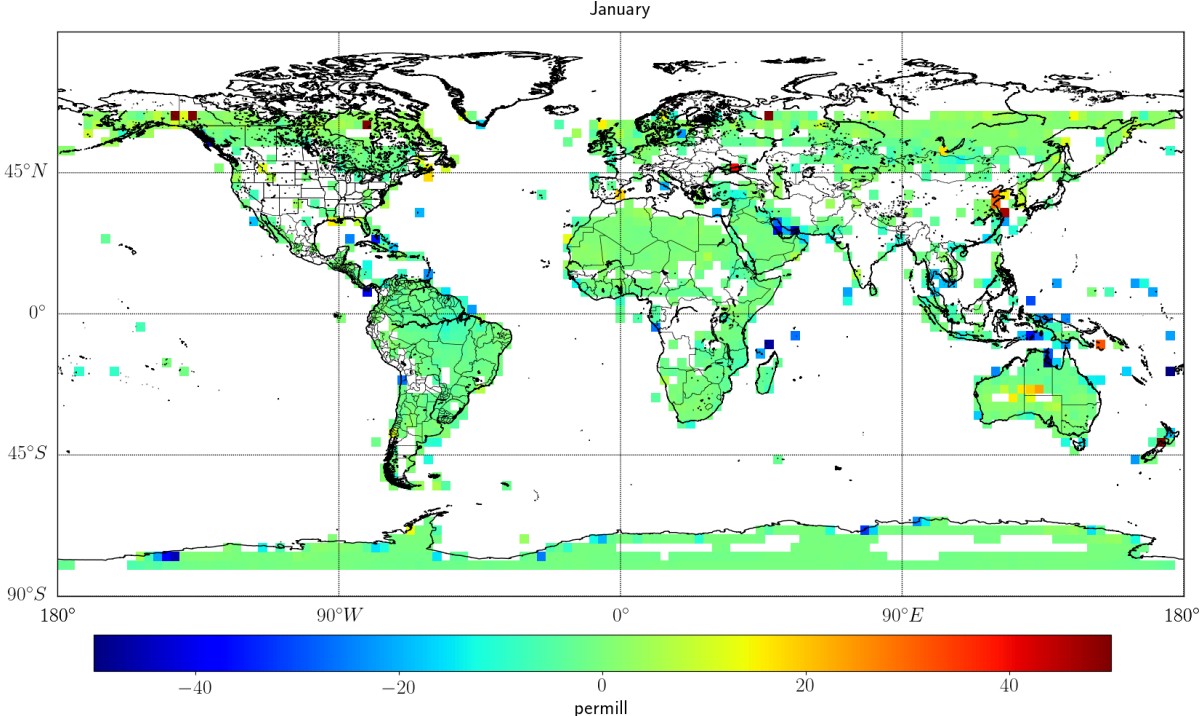

**Figure 18.** As Fig. 16 and Fig. 17, but focused on dual band UVNS.

Figure 18 shows similar results to those for SWIR1 and SWIR3 considered individually. A global systematic bias is present, Hahaof roughly similar magnitude to those shown in Fig. 16 and Fig. 17. However note that some of the localised systematic biases (e.g. Hudson's bay) are no longer apparent. Again this suggests that this global systematic bias is not related to atmospheric scattering.

Within Figs 16, 17 and 18, we can identify several individual retrieval points where there is significant deviation between the scattering and non-scattering cases. These points are clearly linked with the high uncertainty points identified in the error figures Figs 4, 10 and 14, which will cause deviations between the scattering and non-scattering cases.

## 7   Discussion

There are several issues with the assumptions in this study that must be discussed. First, while $\delta^{13}$C ratio use in in situ measurements has been proven many times (Nisbet et al., 2016; Fisher et al., 2017; Rella et al., 2015); it has never been used in total column measurements previously, and there are numerous challenges associated with this. For example the total column as measured by S5P/TROPOMI, S5/UVNS is well mixed above the boundary layer, and therefore will contain

methane advected from different global regions. In addition, the main methane sinks in the atmosphere (OH, OD and Cl) alter fractionation themselves, independent of the methane source type. For example Rigby et al. (2017) associate a fractionation value of 2.6‰ with the Cl sink. Hence, carefully prepared $\delta^{13}C$ databases such as Sherwood et al. (2016) may be not fully relevant to total column measurements. This is a lesser problem while $^{13}CH_4$ retrievals from satellites remain imprecise, but it is not difficult to envisage more advanced future technology surmounting the challenges shown in this work, at which point the use of the $\delta^{13}C$ ratio will have to be revisited, when considering the total column or even limb soundings. For example unique $\delta^{13}C$ ratio values could be assessed, depending on which portion of the atmosphere is considered.

Even if we assume that total column retrievals of $\delta^{13}C$ are not directly comparable to the previously identified $\delta^{13}C$ databases. Retrieving the $\delta^{13}C$ ratio with a 1‰ uncertainty may allow for assessment of how a particular source region changes year on year. This 1‰ value is based on the flask assessments of (Nisbet et al., 2016) and the airborne measurements of (Fisher et al., 2017). However, variations in the $\delta^{13}C$ value of a total column in not directly comparable to these surface and airborne measurements, given that the changes are unlikely to be repeated in the upper atmosphere, and therefore the changes will be dampened. It is therefore likely that higher precision requirements will be required in order to comment on the total column changes, however since no such measurements have been made, it is difficult to say with any certainty what the requirements should be.

Even though we do not assess the accuracy of the spectroscopy of methane isotopologues in this study, we believe that this is necessary for future studies, since minor systematic errors can have a significant impact on the calculated $\delta^{13}C$ ratio. Potential examples of such studies can be found in Galli et al. (2012) and Checa-Garcia et al. (2015). With regards to the HITRAN2012 database, Brown et al. (2013) note that the $^{13}CH_4$ spectral lines used in this study were all measured empirically (i.e captured from in situ/laboratory studies and not assigned by quantum mechanical calculations), and still retain significant levels of uncertainty, especially in relation to atmospheric broadening. The recent spectral line database SEOM-IAS designed for the SWIR3 band on TROPOMI (Birk et al., 2017) shows the benefit of applying non-Voigt broadening profiles to the TROPOMI spectral band, and emphasises the importance of getting the spectroscopy correct, especially for minor species such as $^{13}CH_4$. We therefore emphasise the importance of a full assessment of the spectroscopy of the isotopologue lines, before performing full retrievals.

Comparisons of the total uncertainties described in the main body of the text, and the precision errors shown in Appendix A suggest that systematic errors make up a significant percentage of desired error requirement. We state in the main body of the text that it is typically only random errors that can be reduced through spatio-temporal averaging. However it is important to note that a portion of the represented systematic errors will be pseudo errors, and may well be mitigated by spatio-temporal averaging.

Section 2.1 describes how the Sentinel 5P and Sentinel 5 missions are on different orbits, with S5P having a 13:30 local time in descending node, while S5 has a 09:30 crossing. This means that the synthetic ensemble used in this study (designed for S5P) is not fully representative of the conditions that S5 will observe. It is likely that in reality the solar zenith angles will be higher, and therefore the SNR of all S5/UVNS retrievals will be lower than represented in this study. However, while this means that the errors shown in sects 3.4 and 5.3 will be higher, it is unlikely that any of the conclusions in this paper will

change substantially. In addition, fewer clouds will be present in the morning orbit and therefore S5/UVNS will make more clear sky measurements than S5P/TROPOMI.

Section 5 shows the results from attempting dual band retrievals of $^{13}CH_4$. In reality, a dual band retrieval is likely to introduce additional errors not present in single band retrievals. For example, detector mis-alignment may require additional processing to co-register the images from different bands, through which co-location errors can creep into the process (Worden et al., 2015). However, we do not consider these in this study.

The third most common methane isotopologue is $CH_3D$, making up approximately 0.06% of atmospheric methane. Like $^{13}CH_4$, the ratio of this isotopologue to the main methane concentration can be used to differentiate between methane sources (Rigby et al., 2012). We attempted retrievals of this molecule with RemoTeC (for each of the bands considered in this paper), but were unsuccessful. Most likely the spectral lines present in the HITRAN2012 database are so rare that the retrieval procedure were unable to obtain any information above the noise limit.

Although this study is based on the use of the L-curve method to calculate the regularisation parameter for the Philips-Tikhonov method. RemoTeC can also perform retrievals using a single static parameter. We compared the results of the retrievals from the L-curve method and the static value, in order to identify any points of divergence between the methods, and found no difference in results.

The assessment of scattering scenarios vs non-scattering scenarios in this paper show that calculating the $\delta^{13}C$ value effectively removes the impact of any scattering induced errors. The systematic errors indicated in Figs 16, 17 and 18 suggest that they are unrelated to aerosol scattering, given that large difference do not occur in well known aerosol regions such as the Sahara desert. Unfortunately it is difficult to exactly explain what cause some of the differences observed in Figs 16, 17 and 18, since systematic errors are difficult to quantify (Houweling et al., 2014). However, given that the bias is largely globally consistent, it will not be difficult to apply a correction factor and remove this bias for the differences in the scattering/non-scattering scenarios. As explored above, other systematic errors will be present in the retrievals, and these cannot be simply removed through a global adjustment.

In this study we consider $^{13}CH_4$ and $^{12}CH_4$ as separate targets to be retrieved. What is not considered in this study is jointly retrieving these species, such that the final retrieved VMR is constrained to fit within a prescribed $\delta^{13}C$ range. Such techniques are exhibited by (Worden et al., 2006) when applied to $H_2O$ and HDO ratios. It may be an interesting exercise to undertake a similar investigation using these constraint methods, however when retrieving similar species with GOSAT, (Boesch et al., 2013) state that constricting the solution too much may lead to a false result.

## 8 Conclusions

This study used the well established information content analysis techniques to determine the potential for $^{13}CH_4$ retrievals (and consequently, the $\delta^{13}C$ metric), from the SWIR channels of the current S5P/TROPOMI instrument (2305-2385 nm), and the future S5/UNVS instrument (1590-1675 nm & 2305-2385 nm), assuming clear sky, non-scattering conditions. We used the RemoTeC retrieval software, which is based on a Phillips-Tikhonov regularisation scheme, a synthetic database of over

10k simulated measurements which simulate global atmospheric and surface scenes which S5P/TROPOMI and S5/UVNS will be expected to encounter, and the HITRAN2012 spectroscopic database. For the TROPOMI SWIR3 channel, we find that total uncertainty (for all retrievals with DFS values > 1) has a global mean value of 1 ppb, for the Sentinel 5/UVNS SWIR1 channel, the global uncertainty has a mean value of 0.68 ppb, and a dual band retrieval of both channels has an uncertainty of 0.6 ppb. The SWIR3 shows the poorest performance, with only roughly 50% of the synthetic retrievals passing the DFS > 1 requirement, with forested scenes and high latitude scenes largely filtered out. The SWIR1 and dual band retrievals show a roughly 80% pass rate, with similar magnitudes in error and number of valid retrievals, suggesting that dual band retrievals are dominated by the SWIR1 band. These errors have the potential to be sufficiently low such that the target uncertainty of 0.02 ppb (in order to achieve a $\delta^{13}$C uncertainty of 1‰) can be achieved with spatio-temporal averaging (typically 1 year, if assuming repeat overpasses on a daily basis over a wide region). We also investigate the potential systematic bias effects of uncertainties in the a priori state vector (methane, water vapour), and ancillary information (temperature and pressure profiles), and instrumentation errors on retrievals of $^{13}$CH$_4$ and $\delta^{13}$C. Uncertainty in a priori knowledge of methane and water vapour profile are found to have minimal effects on retrieved results, but uncertainty in temperature and pressure ancillary information lead very large systematic bias effects (primarily on SWIR3 (>60‰), but also significant in SWIR1 (>30‰). Thus, in order to leverage methane isotopologue measurements from S5P/TROPOMI and/or S5/UVNS, better knowledge of the ancillary information is required. Finally we assessed the potential impact of light path induced errors due to scattering vs non-scattering retrievals of $^{13}$CH$_4$ and $^{12}$CH$_4$. Upon comparing the $\delta^{13}$C values for both scattering and non-scattering cases, we found that there are no systematic biases present that could be caused by scattering. We therefore concluded that scattering is not important in calculating the $\delta^{13}$C in any of the TROPOMI/UVNS SWIR bands.

In summary there is limited benefit to attempting the retrieval of $^{13}$CH$_4$ using S5P/TROPOMI at this time. However, the results in this paper suggest that there may be benefits to retrievals of $^{13}$CH$_4$ using the future S5/UVNS instrument, and we encourage research into this area.

*Code and data availability.* The RemoTeC algorithm and synthetic scenario database are available upon discussion with Jochen Landgraf at SRON, all code used to analyse the output from RemoTeC is available upon request from the primary author. The HITRAN spectral line lists are available from hitran.org.

## Appendix A: Precision Errors

In addition to the total uncertainty maps presented in the main text above, this appendix outlines the precision errors associated methane isotopologue retrievals. Allowing for an assessment of how much error can be reduced through spatio-temporal averaging, and what cannot.

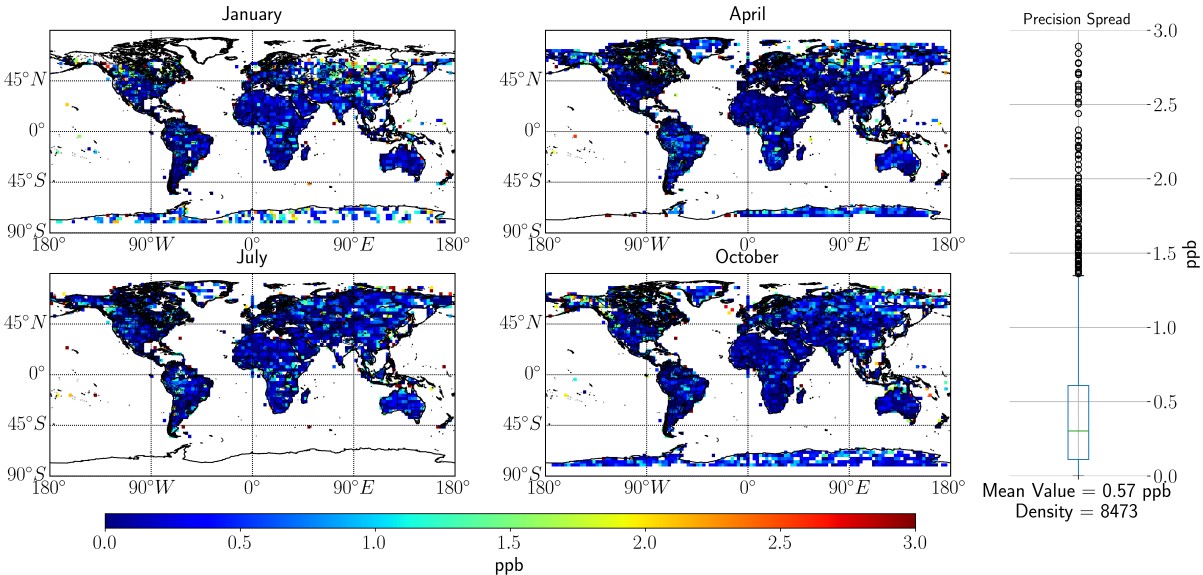

**Figure A1.** Global spread of precision errors based on the RemoTeC synthetic data ensemble for the SWIR1 band, with $^{13}CH_4$ as the target for retrievals. The four main seasons in the synthetic database are represented in this figure by one day in the months of January, April, July and October. The far right panel in the figure outlines the spread of error values over the entire dataset, with the mean value, and the total number of measurements indicated at the bottom.

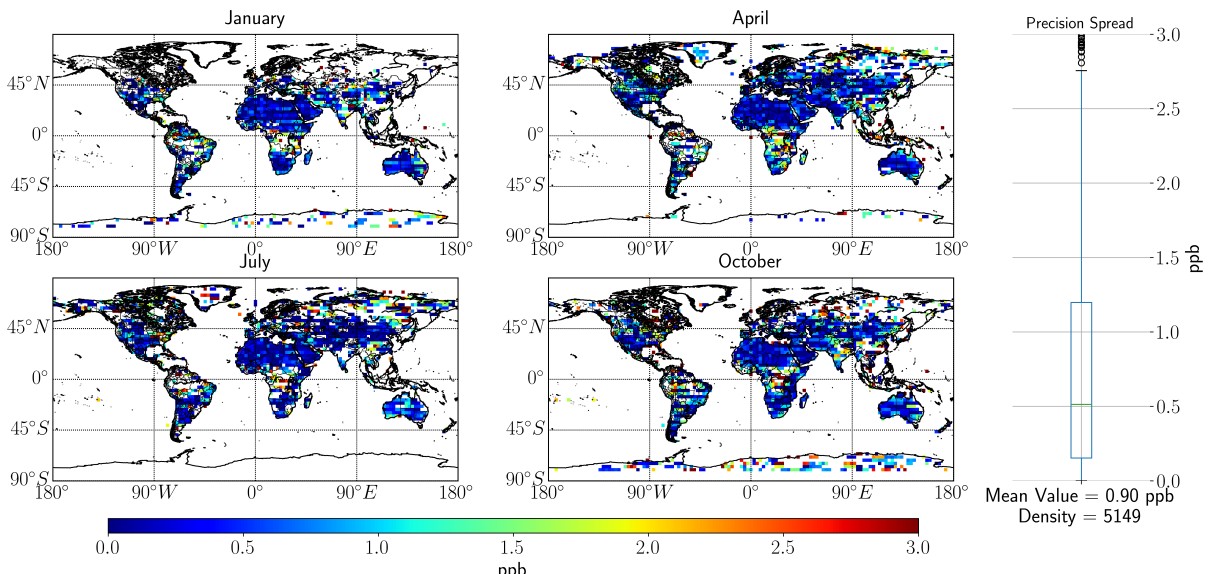

**Figure A2.** As Fig A1, but focused on the SWIR3 band.

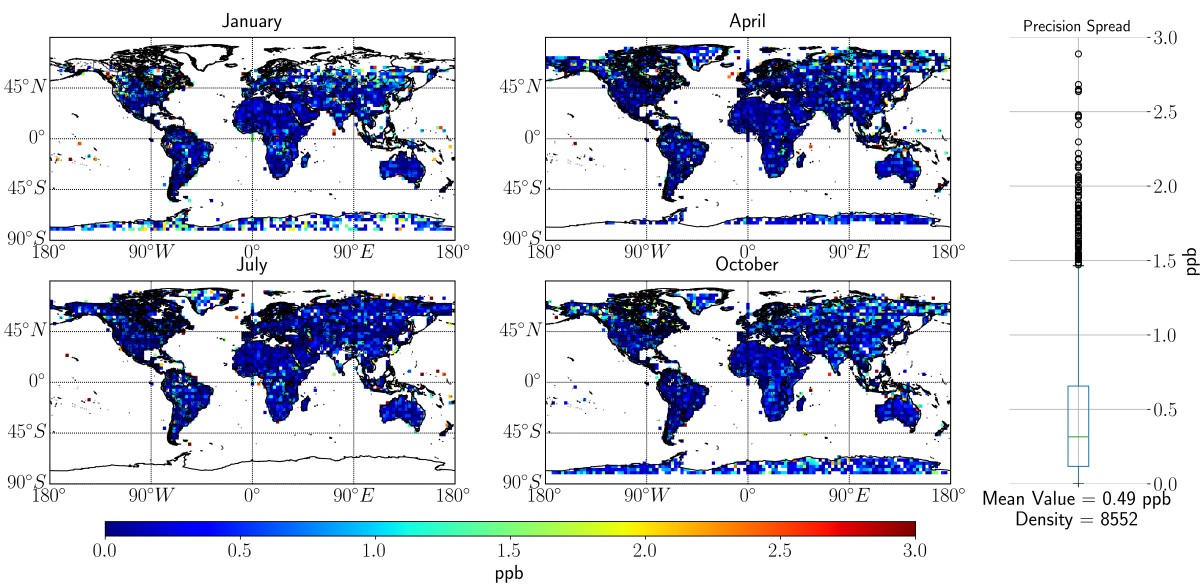

**Figure A3.** As Fig A1, but focused on dual band retrievals.

*Author contributions.* H.H and J.L developed the RemoTeC algorithm and provided aid on its use. E.M performed the analysis and wrote the paper. B.V, H.H and J.L consulted on the interpretation of the results.

*Competing interests.* We declare no competing interests.

*Acknowledgements.* This study has been performed in the framework of the postdoctoral Research Fellowship program of the European Space Agency (ESA).

Thanks to Thorsten Fehr and Alex Hoffmann of ESA for reviewing the paper.

5    Thanks to the reviewers in the peer-review process whose suggestions helped improve the paper. Particularly notable contributions came from Julia Marshall and Thomas Rockmann, whose suggestions resulted in the revision of the original $\delta^{13}$C target requirement of 10‰ to the current 1‰.

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
