# Peer review of "A study of synthetic $^{13}\mathrm{CH}_4$ retrievals from TROPOMI and Sentinel 5/UVNS"

_Atmospheric Measurement Techniques, 2018_

## Short Comment (SC1) · 25 Jan 2019

In this manuscript the authors investigate whether new and future satellite instruments have sufficient precision to provide scientifically valuable data on the isotopic composition of tropospheric methane (CH4). Unfortunately, the precision requirement for isotope information from satellite retrievals assumed in the study (10 per mill) is inadequate for scientific interpretation. The choice of this requirement goes back to Malina et al. (2018) , explained in Figure 1 there, but this figure and interpretation are misleading.

Although the range of isotope source signatures is correct, a satellite instrument will never observe CH4 from a pure source, but only an elevation above a relativley high

background. The (larger) elevations in total column CH4 from TROPOMI are of the order of 100 ppb (for strong elevations) so about 5% above background. Even if this additional CH4 comes from a single source with an isotope signature that is strongly different from the background d13C value of -47 per mill (e.g. biogenic CH4 may be 30 per mill depleted), this would only change the d13C value of the total column by 5% * 30 per mill, thus 1.5 per mill. This is the order of magnitude of isotope variations that was suggested before by isotope specialists as minimum target precision, e.g. by Nisbet et al. (2016), cited in the manuscript. Isotope signals of 10 per mill as assumed in the present manuscript and by Malina et al. (2018) are usually not even observed with in-situ techniques in the boundary layer, unless measurements are performed directly in the plume of a huge local source (e.g. Röckmann et al., 2016, Zazzeri et al., 2016).

The comparison to the way too loose requirements from Malina et al. (2018) leads to the misleading conclusion that that isotope retrievals from space with scientifically relevant precision are within reach for tropospheric CH4. Using a realistic precision requirement, an adequate conclusion would be that scientifically valuable isotope retrievals for CH4 are beyond the performance limits of current and planned instruments.

References:

Malina, E., et al. Information content analysis: The potential for methane isotopologue retrieval from GOSAT-2, Atmos. Meas. Tech., 11, 1159–1179, doi:10.5194/amt-11-1159-2018, 2018.

Nisbet, E. G., et al. Rising atmospheric methane: 2007–2014, growth and isotopic shift, Global Biogeochem. Cycles, 30, doi:10.1002/2016GB005406, 2016.

Röckmann, T. et al. In situ observations of the isotopic composition of methane at the Cabauw tall tower site. Atmos. Chem. Phys. 16, 10469-10487, doi:10.5194/acp-16-10469-2016, 2016.

Zazzeri, G. et al. Carbon isotopic signature of coal-derived methane emissions to the

atmosphere: from coalification to alteration. Atmos. Chem. Phys. 16, 13669-13680, doi:10.5194/acp-16-13669-2016, 2016.

---

## Author Comment (AC1) · 12 Mar 2019

Dear Professor Roeckmann,

Thank you for your comment. Our assessment of your comment is that our basic assumption of a d13C requirement of 10 per mil uncertainty is insufficient to differentiate between methane source types, based on the fact that would be impossible to differentiate unique source from the tropospheric background d13C.

Our terminology in referring to sources was inaccurate in our manuscript. This created the wrong impression that we claim to be able to differentiate between pure sources.

In fact, we do acknowledge that this is very unlikely. Rather, what we are hoping to achieve with Sentinel 5 (we conclude that Sentinel 5P is not currently suited to this task) is regional source type differentiation. For example (Fisher et al., 2017) show that wetland emissions from boreal regions have an integrated signature of -70 per mil, which were identified using aircraft over wide tracks of land. We therefore think that using S5 we may be able to distinguish between a wetland region, and for example a biomass burning region with this requirement of 10 per mil. Fisher et al. suggest that regional isotopic signatures could be incorporated into global and regional models. These models could then be compared against S5 measurements of d13C.

We will therefore clarify our requirements accordingly.

Fisher, R. E. et al. (2017) 'Measurement of the 13C isotopic signature of methane emissions from northern European wetlands', Global Biogeochemical Cycles, 31(3), pp. 605–623. doi: 10.1002/2016GB005504.

---

## Referee Comment (RC1) · Anonymous Referee #1 · 18 Mar 2019

This manuscript by Malina at al. examines whether S5P/TROPOMI and S5 can retrieve 13CH4 with sufficient precision to differentiate CH4 source contributions. Carbon isotope measurement in CH4 could provide useful information on the atmospheric CH4 budget and the changing CH4 sources/sinks that are not fully understood. Therefore, data availability of 13CH4 measurements are highly anticipated but challenging for carbon cycle science community.

The information Content (IC) analysis approach in the study is similar to the one presented by Malina et al. (ATM, 2018) for GOSAT-2. Both are feasibility tests to these new/future satellites. Both studies applied IC analysis to the synthetic measurements,

assuming clear sky condition (non scattering atmosphere). One of the main differences between GOSAT-2 and S5P/S5 is resolution due the different spectrometers (resolutions), FTS (0.2 cm-1) and push broom spectrometer (0.45 cm-1).

The paper would be interesting for the readers of Atmospheric Measurement Techniques, in particular for those studying GHG from the space. This kind of feasibility test is essential before handling real data and contribute to exploring the possibilities in coming satellites. However, the paper needs to be improved. There are missing information and editorial errors (missing words, incomplete sentences, and so on), and inconsistencies with figures. Some figures are not clear.

The paper potentially contributes to the GHG community. However, at this stage, I recommend major modification for further consideration.

Major comments

Overall, the manuscript has a lack of information and consistency in figures. Please check the figures (including captions) and descriptions in the main text. Figures should be self-descriptive. Also, the figure legends should not overlap/distract the plots (see more in the specific comments).

In Introduction, the authors said 'the key questions becomes whether SNR or spectral resolutions is the key limiting factor in the retrieval of methane isotopologues". At the end of Sect. 4.5, the authors conclude that "SNR is more important... than spectral resolution". However, this comes from the comparison between SWIR1 and SWIR3. SWIR1 has higher SNR than SWIR3, but they both have the same spectral resolution, which is lower than GOSAT-2. Before the conclusion, a discussion on spectral resolution vs SNR should be given.

Definition of latitude bands (low-latitude, mid-latitude and, high-latitude) should be given before the results are presented (from Section 3 onward). Alternatively, specific regions should be referred to. Otherwise, readers might be lost. For example, in lowand mid-latitudes, the regional differences (between desert areas and non-temperate areas, inland and islands, etc) are more evident than the latitudinal characteristics.

Specific comments

- Page 7, L4 'This paper builds on this study'. These two "this" must refer to different studies. Please rephrase this sentence.

- Page 9, L7: "The spectral fit quality is good, with a chi-square value equal to 1". How was the chi-square value of 1 derived? Fig. 1 reads chi-square of 1.15, not 1.

- Page 10, L1: "disagree with the 'truth', notably the methane lines at 1670 nm". However, no such disagreement is seen in the top panel in Fig. 1.

- Page 10 L3: Please specify more of "complex behaviour" about the cause of difference?

- Figure 1: Caption reads "-1.4S, -47.81W for a day in January 2015". However the middle panel shows "Measured, 16/7/2015, 51.63 39.39". They should be consistent. No right-hand scale in the bottom panel, which is mentioned in the caption.

- Figure 2: Overlapped legends are destructive. Please move them outside the panels (same for Figs. 8 and 12). Four colours indicate different regions, but it is hard to distinguish them. It would be more informative if the legends include the representing regions (not only latitude-longitude information).

- Figure 3: To see the overall performance, it would be helpful to have a total number of the measurement (before measurements with DFS (<1) have been filtered out).

- Page 11, L15: 'almost 3000 additional valid retrievals, which is roughly 30% of the ensemble'? This statement is not clear. What do the authors mean by "3000 additional"?

- Figure 4: 'lobal' should be 'Global'?

- Page 12, L11: what 'show'? The subject of the sentence is missing.

- Page 12, L13: the last sentence seems incomplete.

- Table 3: The units should be given for 13CH4 and d13C intercept and sigma, separately (same for Table 4).

- Figures 5, 6 and 11: Please give the units for x-axis and y-axis. Also, for Figs 5 and 6, please describe in the caption which parameters perturbed are.

- Figure 6: Is the caption correct?

- Page 16, L29: ILSF should be spelled out.

- Page 22, L4: SWIR3 should be SWIR1?

- Page 23, L1: "Radiometric offset errors are not significant in the SWIR3 as opposed to the SWIR1 band'. More explanation and possible reasons for or discussion on this difference should be given.

- Page 27, L34: What is "the L-curve method"? It should be introduced before being mentioned in the Discussion section.

---

## Referee Comment (RC2) · Anonymous Referee #2 · 29 Mar 2019

The manuscript presented a study on retrievals of 13CH4 from TROPOMI and Sentinel 5/UVNS. It is well written and very informative. I suggest it be accepted from publication after minor revision.

Major comments:

1. Although reference papers are provided, I think it is helpful for the reader if the authors can provide a clearer description of the remoTeC algorithm, for example, the components of the state vector etc.

2. More explanations about why the average kernel for 13CH4 is different from 12CH4 are also welcome.

Minor comments:

1. Line 21, Page 1: 'The disagreement ...' The bottom-up approaches have large uncertainty as well.

2. Line 22, Page1: 'or incorrect transport ...', There also are large uncertainties in modelling CH4 chemical losses.

3. Line 15, Page 3: 'Parker et al.,...', Works by Frankenberg et al., 2005 and 2011 should also be cited.

4. Line 10, Page 5: A comparison of 13CH4 and 12CH4 absorptions at different atmosphere levels can be useful for the reader to understand the different sensitivity of the TROMOPI instrument to their abundance.

5. Line 28, Page 6: '...and that is potential...', The whole sentence is not clear.

6. Fig 1: no unit shown for Jacobian. Also, no right-hand scale for 12CH4.

7. Line 5, '...errors in Figure 4..'. Some explanation about the spots with high uncertainty (>1.5 ppb) will be helpful

---

## Author Comment (AC3) · 31 May 2019

Dear Reviewer,

Thank you for your review of our paper. In order to respond we have kept your original comments in the black text below, our responses are in blue, and our proposed changes to the paper are in underlined blue text.

The manuscript presented a study on retrievals of 13CH4 from TROPOMI and Sentinel 5/UVNS. It is well written and very informative. I suggest it be accepted from publication after minor revision.

Major comments:

1. Although reference papers are provided, I think it is helpful for the reader if the authors can provide a clearer description of the remoTeC algorithm, for example, the components of the state vector etc.

We have now included additional details about the RemoTeC algorithm, including the state vector, and the cost function that is minimized. Please see the updated section 2.2.

2. More explanations about why the average kernel for 13CH4 is different from 12CH4 are also welcome.

We have included the following statement in section 3.2. Which we believe answers the question as to the difference between the averaging kernels.

The $^{12}CH_4$ cAK exhibits the typical behaviour of $CH_4$ cAKs (e.g. Hu et al. (2016), which is expected since $^{12}CH_4$ makes up 98% of atmospheric $CH_4$. However the $^{13}CH_4$ cAK exhibits behavior closer to that of CO (Landgraf et al. 2016). Given that $^{13}CH_4$ makes up ~1.1% of atmospheric $CH_4$, the retrieval column loses sensitivity in the lower atmosphere, where $H_2O$ dominates. Borsdorff et al. (2014) show that in the case where sensitivity is low in the troposphere, the cAK values are enhanced at other altitudes. This is apparent in the cAKs of $^{13}CH_4$ in Fig. 2, where cAK values larger than those of $^{12}CH_4$ are observed.

Minor comments:

1.    Line 21, Page 1: 'The disagreement ...' The bottom-up approaches have large uncertainty as well.

We agree, we have modified the sentence to read as follows:

This disagreement is likely due to currently limited observations, incorrect atmospheric transport assumptions, uncertainties associated with bottom up inventories and uncertainties in modelling $CH_4$ chemical losses.

2. Line 22, Page1: 'or incorrect transport ...', There also are large uncertainties in modelling CH4 chemical losses.

Agreed, we have modified this sentence to include this statement. Please see the modification to point 1 above.

3. Line 15, Page 3: 'Parker et al.,...', Works by Frankenberg et al., 2005 and 2011 should also be cited.

Thank you, we have inserted these references.

4. Line 10, Page 5: A comparison of 13CH4 and 12CH4 absorptions at different atmosphere levels can be useful for the reader to understand the different sensitivity of the TROMOPI instrument to their abundance.

A discussion of the variation of Jacobians w.r.t. 13CH4 and 12CH4 is given in Malina et al. 2018. We have pointed to this work by inserting the following into the Jacobian bullet point in section 2.5.

In this study we investigate how the total column Jacobians vary between the isotopologues, however Malina et al. (2018} give examples of how the Jacobians vary on a profile basis.

5. Line 28, Page 6: '...and that is potential...', The whole sentence is not clear.

We have re-written the paragraph containing this, based on your comment, and based on the short comment of Professor Roeckmann as follows:

Malina et al. (2018) identify a target total uncertainty for d13C of 10‰ as a more realistic and potentially achievable value (based on simulations with GOSAT-2). Recently Fisher et al. (2017) show that a distinct regional d13C signature can be measured, in their particular case for boreal forest regions. Therefore, as opposed to tracking d13C changes, we may be able to identify the source type of regional methane sources on a global scale, thus adding additional information to the top down methane budget.

6. Fig 1: no unit shown for Jacobian. Also, no right-hand scale for 12CH4.

Jacobian units have been added to Figures 1 and 7. The caption for Figure 1 was incorrectly labeled, there should be no 'right-hand scale or left-hand scale' in the caption, we have therefore removed these. Please note that Reviewer 1 found that the coordinates and date given in the caption do not match those shown in the legend of the middle panel. The caption has been updated to reflect this.

7. Line 5, '...errors in Figure 4..'. Some explanation about the spots with high uncertainty (>1.5 ppb) will be helpful

The retrievals with high errors are characterized by low SNR retrievals. We have inserted the following sentence into the document at the end of this paragraph associated with Figure 4.

An investigation showed that these retrievals are all captured under low SNR conditions, largely driven by SZA and albedo, thus leading to high uncertainty.

Please note that in addition to the changes indicated above in response to your comments, we have also made changes to additional editorials we spotted. In addition, in response to Thomas Roeckmann's criticism, we have included a short section on the effects of scattering on the retrievals of $^{13}CH_4$. This now forms section 6 of the paper. Additional necessary details on the scattering elements of RemoTeC have been included in the RemoTeC section.

Please note that we spotted errors in Sections 3.5 and 4.5 of our original submission. The results shown in Tables 3, 4 and Figures 5, 6 and 11 were generated without the DFS > 1 filter that were included for the maps plots present in the rest of our submission. We have reprocessed this data, and have updated

the relevant figures and tables, including the filtering criteria. We have updated the relevant portions of the text that reference the original results.

---

## Author Comment (AC4) · 16 Sep 2019

Further to our initial response, upon further reviews we have been convinced that a requirement of 10 per mil is too large to be of any significant benefit to the modelling or instrument communities. Based on the recommendations of one of the reviewers, we have increased our requirements to 1 per mil. Nisbet et al (2016) show that inter annual variations of d13C can be as much as 1 per mil, and we hope that this could be captured with a total column uncertainty of 1 per mil.

Nisbet, E. G., Dlugokencky, E. J., Manning, M. R., Lowry, D., Fisher, R. E., France, J. L., Michel, S. E., Miller, J. B., White, J. W. C.,Vaughn, B., Bousquet, P., Pyle, J. A.,

Warwick, N. J., Cain, M., Brownlow, R., Zazzeri, G., Lanoisellé, M., Manning, A. C., Gloor, E.,Worthy, D. E. J., Brunke, E.-G., Labuschagne, C., Wolff, E. W., and Ganesan, A. L.: Rising atmospheric methane: 2007-2014 growth and isotopic shift, Global Biogeochemical Cycles, 30, 1356–1370, https://doi.org/10.1002/2016GB005406, 2016.

---

## Author Comment (AC2)

Dear Reviewer,

Thank you for your review of our paper. In order to respond we have kept your original comments in the black text below, our responses are in blue, and our changes to the paper are in underlined blue text.

This manuscript by Malina at al. examines whether S5P/TROPOMI and S5 can retrieve 13CH4 with sufficient precision to differentiate CH4 source contributions. Carbon isotope measurement in CH4 could provide useful information on the atmospheric CH4 budget and the changing CH4 sources/sinks that are not fully understood. Therefore, data availability of 13CH4 measurements are highly anticipated but challenging for carbon cycle science community.

The information Content (IC) analysis approach in the study is similar to the one presented by Malina et al. (ATM, 2018) for GOSAT-2. Both are feasibility tests to these new/future satellites. Both studies applied IC analysis to the synthetic measurements, assuming clear sky condition (non scattering atmosphere). One of the main differences between GOSAT-2 and S5P/S5 is resolution due the different spectrometers (resolutions), FTS (0.2 cm-1) and push broom spectrometer (0.45 cm-1).

The paper would be interesting for the readers of Atmospheric Measurement Techniques, in particular for those studying GHG from the space. This kind of feasibility test is essential before handling real data and contribute to exploring the possibilities in coming satellites. However, the paper needs to be improved. There are missing information and editorial errors (missing words, incomplete sentences, and so on), and inconsistencies with figures. Some figures are not clear.

The paper potentially contributes to the GHG community. However, at this stage, I recommend major modification for further consideration.

Major comments

1. Overall, the manuscript has a lack of information and consistency in figures. Please check the figures (including captions) and descriptions in the main text. Figures should be self-descriptive. Also, the figure legends should not overlap/distract the plots (see more in the specific comments).

   Thank you for your comments, we have addressed your concerns in the specific comments below.

2. In Introduction, the authors said 'the key questions becomes whether SNR or spectral resolutions is the key limiting factor in the retrieval of methane isotopologues". At the end of Sect. 4.5, the authors conclude that "SNR is more important... than spectral resolution". However, this comes from the comparison between SWIR1 and SWIR3. SWIR1 has higher SNR than SWIR3, but they both have the same spectral resolution, which is lower than GOSAT-2. Before the conclusion, a discussion on spectral resolution vs SNR should be given.

   Thank you for your point, however we respectfully disagree with the need for a discussion on spectral resolution vs SNR. This is a topic that has been extensively discussed elsewhere (e.g.

(Wunch *et al.*, 2011), and we do not feel that this is a controversial point. The key reasons for the difference between the SWIR1 and SWIR3 SNRs are the solar irradiance and surface albedo, both of which are lower in the SWIR3 as opposed to the SWIR1. This will lead to a lower SNR irrespective of spectral resolution. To emphasis this point, we have entered the following sentence into section 4.6.

The key reasons for the lower SNR in the SWIR3 band as opposed to SWIR1 is the lower solar irradiance and surface albedo at these wavelengths.

3. Definition of latitude bands (low-latitude, mid-latitude and, high-latitude) should be given before the results are presented (from Section 3 onward). Alternatively, specific regions should be referred to. Otherwise, readers might be lost. For example, in and mid-latitudes, the regional differences (between desert areas and non-temperate areas, inland and islands, etc) are more evident than the latitudinal characteristics.

We agree, and we have inserted the following sentences in to section 2.3 to provide a clear definition of high-latitude, etc.

Through this paper, we will refer to "latitudinal bands", which we split in three distinct areas. Tropical (~0-20°), mid-latitude (~20-60°) and high latitude (>60°). These are typically how model atmospheres are split (e.g. mid-latitude summer etc). Surface conditions will cause the results in this bands to vary, and we identify and significant regions that show significant deviation.

In the case of where regional differences being of high importance, we agree on this point. In our original submission, we typically highlighted specific areas that showed clear differences from other areas in a similar latitude band. For example highlighting the Amazon rain forest, or desert scenes. We therefore believe that we do not need to emphasis this point any further.

Specific comments

4. - Page 7, L4 'This paper builds on this study'. These two "this" must refer to different studies. Please rephrase this sentence.

Thank you, we have now re-written this sentence as follows:

"This paper builds on Malina et al. (2018)"

5. - Page 9, L7: "The spectral fit quality is good, with a chi-square value equal to 1". How was the chi-square value of 1 derived? Fig. 1 reads chi-square of 1.15, not 1.

We use the method outlined in (Galli *et al.*, 2012), and we have now included this reference in section 2.5, where the assessment statistics are described. We have also changed the chi-squared value reference in the paragraph below figure 1.

6. - Page 10, L1: "disagree with the 'truth', notably the methane lines at 1670 nm". However, no such disagreement is seen in the top panel in Fig. 1.

Agreed, we have looked further into this and found that the disagreement is more random in nature, rather than a persistent feature as originally thought. We have therefore modified this sentence to read.

However, there are some points which could be interpreted as not due to random noise, where the retrieval seems to disagree with the 'truth', notably the methane lines between 1645 and 1650 nm. However, upon further investigation we found that these features do not consistently appear in spectral residuals. Therefore the disagreements shown in Fig. 1 are random in nature.

7. - Page 10 L3: Please specify more of "complex behaviour" about the cause of difference?

This is now redundant given the changes outlined in point 6 above.

8. - Figure 1: Caption reads "-1.4S, -47.81W for a day in January 2015". However the middle panel shows "Measured, 16/7/2015, 51.63 39.39". They should be consistent. No right-hand scale in the bottom panel, which is mentioned in the caption.

This figure had been updated prior to submission, but unfortunately, the caption was not changed to reflect this. The figure has now been updated so that the legend and the caption match. References to left-hand scale and right hand scale have been removed as these are no longer relevant. Figure 7 has also been updated in this fashion.

9. - Figure 2: Overlapped legends are destructive. Please move them outside the panels (same for Figs. 8 and 12). Four colours indicate different regions, but it is hard to distinguish them. It would be more informative if the legends include the representing regions (not only latitude-longitude information).

Thank you, we have now moved the legends so they do not overlap with the plots. In addition we have highlighted the representative region for each example, e.g. Tropical or Mid-latitude etc. This has been performed for each of the Averaging Kernel plots.

10. - Figure 3: To see the overall performance, it would be helpful to have a total number of the measurement (before measurements with DFS (<1) have been filtered out.

In the original submission we state in section 2.3 that there roughly 10000 simulated synthetic spectra. We have now modified this to give the exact number (11041). This now gives context to the retrieval numbers stated in Figure 3, and other related figures.

11. - Page 11, L15: 'almost 3000 additional valid retrievals, which is roughly 30% of the ensemble'? This statement is not clear. What do the authors mean by "3000 additional"?

We agree, and we have modified this sentence to read as follows:

The differences in measurement densities indicated in both Figs 3 and 9, show that the SWIR1 band has almost 3000 additional valid retrievals that pass the filtering criteria as opposed to SWIR3. These additional valid retrievals represent roughly 30% of the total ensemble, and is therefore a significant proportion.

We have also moved this sentence to section 4.3.

12. - Figure 4: 'lobal' should be 'Global'?

Thank you, updated.

13. - Page 12, L11: what 'show'? The subject of the sentence is missing.

Fixed.

14. - Page 12, L13: the last sentence seems incomplete.

We have modified the sentence to read as follows:

Meaning that a target uncertainty of 0.1 ppb is a more accurate requirement, as opposed to the original goal of 0.2 ppb.

15. - Table 3: The units should be given for 13CH4 and d13C intercept and sigma, separately (same for Table 4).

We have inserted the units "ppb" for the 13CH4 intercept and sigma columns for both Tables 3 and 4, and we have inserted the permil symbol for d13C.

16. - Figures 5, 6 and 11: Please give the units for x-axis and y-axis. Also, for Figs 5 and 6, please describe in the caption which parameters perturbed are.

The units for 13CH4 on these axes has been mode more prominent as compared to the original. The d13C plots have had 'permil' included on the axis. The captions of Figs 5 and 6 have been updated to included details on the parameter perturbation.

17. - Figure 6: Is the caption correct?

Thank you for identifying this. This caption has been updated.

18. - Page 16, L29: ILSF should be spelled out.

The acronym for ILSF had already been identified earlier in the document, page 7, line 6.

19. - Page 22, L4: SWIR3 should be SWIR1?

Thank you, updated.

20. - Page 23, L1: "Radiometric offset errors are not significant in the SWIR3 as opposed to the SWIR1 band'. More explanation and possible reasons for or discussion on this difference should be given.

We have replaced this sentence with the following paragraph to give more details.

Radiometric offset errors are less significant in the SWIR3 as opposed to the SWIR1 band. Radiometric offset errors typically lead to underestimation in the surface reflectance estimate (in the absence of aerosols/clouds) {Kuze et al., 2014). Given that the SWIR1 operates at a higher radiance magnitude than the SWIR3, any minor errors in the surface reflectance estimate will likely lead to larger errors in the calculated radiance, as opposed to the true radiance. Thus leading to larger errors in the SWIR1.

21. - Page 27, L34: What is "the L-curve method"? It should be introduced before being mentioned in the Discussion section.

We have expanded the description of the RemoTeC algorithm in section 2.2, this now includes an introduction to the L-curve method, and a background reference.

Please note that in addition to the changes indicated above in response to your comments, we have also made changes to additional editorials we spotted. In addition, in response to Thomas Roeckmann's criticism, we have included a short section on the effects of scattering on the retrievals of $^{13}CH_4$. This now forms section 6 of the paper. Additional necessary details on the scattering elements of RemoTeC have been included in the RemoTeC section.

Please note that we spotted errors in Sections 3.5 and 4.5 of our original submission. The results shown in Tables 3, 4 and Figures 5, 6 and 11 were generated without the DFS > 1 filter that were included for the maps plots present in the rest of our submission. We have reprocessed this data, and have updated the relevant figures and tables, including the filtering criteria. We have updated the relevant portions of the text that reference the original results.

Galli, A. *et al.* (2012) 'CH4, CO, and H2O spectroscopy for the Sentinel-5 Precursor mission: An assessment with the Total Carbon Column Observing Network measurements', *Atmospheric Measurement Techniques*, 5(6), pp. 1387–1398. doi: 10.5194/amt-5-1387-2012.

Wunch, D. *et al.* (2011) 'The Total Carbon Column Observing Network', *Phil. Trans. R. Soc. A*, 369, pp. 2087–2112. doi: 10.1098/rsta.2010.0240.

---

## Editor Decision (ED1)

**Review of "A study of synthetic $^{13}CH_4$ retrievals from TROPOMI and Sentinel 5/UVNS"**

Julia Marshall

Max Planck Institute for Biogeochemistry
07745 Jena, Germany

The paper is clearly written and structured, and lays out a straightforward analysis of the feasibility of retrieving $^{13}CH_4$ from two different spectral windows given the instrument characteristics and spectral resolution of TROPOMI and Sentinel 5/UVNS.

Unfortunately I think the study is fundamentally flawed in its assumption that a $\partial^{13}C$ uncertainty of 10‰ is sufficient to differentiate source types. This has already been pointed out in the interactive discussion (Röckmann, 2019), which indicated that such large signals are rarely measured *in situ* in the boundary layer, let alone in the total column. The response from the authors referred to a study by Fisher et al. (2017), using airborne measurements above boreal regions, claiming that they measured an integrated signature of -70‰ over wide tracts of land. This is a clear misinterpretation of the findings of Fisher et al. (2017): If one refers to Figure 6 of this paper, it is clear that $\partial^{13}C$ values varied by less than 1‰, between -47.7‰ and -46.7‰. Only by finding the intercept on a Keeling plot (Keeling, 1958) do they deduce that the source which is adding methane to the background has a signature of -70‰. This is never measured directly, outside of chamber measurements.

The authors further suggest that regional isotopic signatures could be incorporated into global and regional models, and that these models could then be compared against S5 measurements of $\partial^{13}C$. My colleague Tonatiuh Nuñez Ramirez has done just this in the course of his doctoral studies (in preparation), and has allowed me to use his modelled $\partial^{13}C$ fields for this review. Using the TM3 transport model, he has used regional isotopic signatures for a range of source categories, drawing upon literature values. He has then performed inversions to adjust the methane fluxes based upon the additional constraint of *in situ* $^{13}CH_4$ measurements. Thus these fields should be representative of realistic variability for both $CH_4$ and $^{13}CH_4$. The resultant $\partial^{13}C$ fields were calculated following Equation 2 of the manuscript being reviewed here.

Figure 1, shows the resultant $\partial^{13}C$ value in the lowest-most model level for a single time step, namely July 1, 2010 at 00 UTC. The model resolution is rather coarse at $\sim 3.8°$ by $5°$, but this is consistent with the spatial averaging that is foreseen by the authors in order to improve the precision of the spaceborne measurement. Note that here the maximum range across the entire globe is less than 5‰, and this is for near-surface values, where the variability can be expected to be the highest. While there is a large range in the isotopic signature of different source processes, the atmosphere is constantly mixing and diluting these signatures with the background values.

[Figure]

**Figure 1.** An instantaneous plot of $\partial^{13}C$ in the model level closest to the surface for July 1, 2010 at 00 UTC.

To see the effect of looking at the column-integrated values, refer to Figure 2, in which the range is less than 2‰. Recall that this is for an instantanous value: when averaged over time, such as the monthly or seasonal scales suggested in the manuscript, the gradients are even further reduced. Also, a flat averaging kernel (i.e. pressure-weighted average) was assumed. Given the lower sensitivity in the lower troposphere presented in the paper, the actual gradients would be even smaller.

[Figure]

**Figure 2.** An instantaneous plot of modelled pressure-weighted column-averaged $\partial^{13}C$ for July 1, 2010 at 00 UTC.

Based on this analysis of the signals expected to be seen in the atmosphere, I find that I am forced to agree with the critical comment of (Röckmann, 2019), and conclude that the precision requirement assumed in the paper is not sufficient for scientific interpretation. Indeed, a precision at least a factor of 10 higher would be required. While it is true that the 10‰ requirement has already been published in (Malina et al., 2018), it does not hold up to scrutiny. As such, I cannot recommend this paper for publication.

Should the editor disagree, and choose to publish the paper as a theoretical exercise in retrieving potential signals larger than those found in the Earth's atmosphere, I have a series of minor comments, outlined below.

**1  Minor comments**

P1, L20: $CO_2$ is not really the "main GHG", but rather the "main anthropogenically-influenced greenhouse gas".

A recurring comment: I'm not sure if it's a typesetting issue, but there are several inconsistencies with nested parentheses, including: P1, L22-23; P2, L8-9; P7, L24-25; P8, L30; P25, L2

The wrong citation form (again with the parentheses) is used in P3, L19.

P3, L32-35: the abbreviation "sect" should be "Sect.", I guess.

P4, L13: add a comma after citation.

P4, L18: In English colons are usually used for denoting times.

P5, L4: I'd hyphenate it as: "36-layer plane-parallel"

P5, L28: I feel like a word is missing, perhaps "spectral dependence of surface albedo"?

P6, L14: identified -> described

P6, L21: data "are"; also P9, L22

P6, L25: The are/will be construction to denote present and future is awkward and should be rewritten.

P7, L4: instrumentin -> instrument in

P8, L28: funny (LaTeX?) quotes on 'truth'; also on P6, L23; P25, L9

P16, L6: should "renders" be "reduces"?

P16, L7: I'd suggest changing "to better knowledge of the ancillary information" to "to improve knowledge of the ancillary information". While "better" can be a verb, here it makes it difficult to parse.

P17, L11: The sentence starting with "While" should be combined with the previous sentence (with a comma).

P21, caption: Should say SWIR3.

P22, L7-8: Reads a bit awkwardly, could be rewritten.

P23, L2: up roughly -> up to roughly

P24, L2: failed convergence -> failed to converge

P24, L10: move comma to after "identify"

P25, L23: comparising -> comparing

P28, L10: makes limited impact -> has a limited impact

P28, L12: remove comma

P30, L4-5: the Hudson bay -> Hudson Bay

**References**

Fisher, R. E., France, J. L., Lowry, D., Lanoisellé, M., Brownlow, R., Pyle, J. A., Cain, M., Warwick, N., Skiba, U. M., Drewer, J., Dinsmore, K. J., Leeson, S. R., Bauguitte, S. J.-B., Wellpott, A., O'Shea, S. J., Allen, G., Gallagher, M. W., Pitt, J., Percival, C. J., Bower, K., George, C., Hayman, G. D., Aalto, T., Lohila, A., Aurela, M., Laurila, T., Crill, P. M., McCalley, C. K., and Nisbet, E. G.: Measurement of the $^{13}$C isotopic signature of methane emissions from northern European wetlands, Global Biogeochemical Cycles, 31, 605–623, https://doi.org/10.1002/2016GB005504, 2017.

Keeling, C. D.: The concentration and isotopic abundances of atmospheric carbon dioxide in rural areas, Geochimica et Cosmochimica Acta, 13, 322 – 334, https://doi.org/https://doi.org/10.1016/0016-7037(58)90033-4, http://www.sciencedirect.com/science/article/pii/0016703758900334, 1958.

Malina, E., Yoshida, Y., Matsunaga, T., and Muller, J.-P.: Information content analysis: the potential for methane isotopologue retrieval from GOSAT-2, Atmospheric Measurement Techniques, 11, 1159–1179, https://doi.org/10.5194/amt-11-1159-2018, https://www.atmos-meas-tech.net/11/1159/2018/, 2018.

Röckmann, T.: Interactive comment on "A study of synthetic $^{13}$CH$_4$ retrievals from TROPOMI and Sentinel 5/UVNS Part 1: non scattering atmosphere" by Edward Malina et al., Atmospheric Measurement Techniques Discussions, https://doi.org/10.5194/amt-2018-450-SC1, 2019.

---

## Author Response (AR2)

Responses to Reviewers and updates.

The reviewer comments are kept in black, author responses are in blue, and changes are underlined blue.

Reviewer 1.

Thanks to this reviewer for a detailed read of our paper.

I would like to suggest to add some more descriptions and discussions regarding the comparison between non-scattering atmosphere and scattering atmosphere.

1. Section 6

What are blank land areas in Figs. 16, 17 and 18? Are there no retrievals available? Or are there no difference between non-scattering and scattering atmosphere?

Blank land areas are due to the retrievals failing to converge and therefore are not included in this analysis.

This is now stated at the beginning of section 6.

In the cases of SWIR1 and SWIR3, the authors say that notable exceptions (pronounced spike) in systematic error over the Hudson Bay in Canada. Besides the Hudson Bay, similar or larger differences appear on other places, such as over tropical Pacific, and the Gulf in the Middle East.

Thank you, we have added this comment.

In the case of SWIR1+SWIE3, the authors say "some of the localized systematic bias (e.g., the Hudson Bay) are no longer apparent". However, one positive spike (>30 permill) appears over the Hudson Bay, while negative spots appear in the first two cases there.

Thank you for pointing this out. This particular location is characterized by high retrieval uncertainties, which causes the solutions for the scattering and non-scattering cases to differ. We have identified this in the text.

Why the global mean biases (differences between non-scattered and scattered) appear to be negative for both SWIR1 and SWIR3? Is the global mean bias negative also in SWIR1+SWIR3?

This is very difficult to explain, given that this phenomenon is not random in nature. As we explain in the discussion, systematic errors are difficult to quantify. In all likelihood, the extreme differences cases seen in Figures 16, 17 and 18 skew the global average towards the negative values, as there are more negative cases than positive cases.

2. Discussion

The authors state in the discussion (P35, L9-10) that "given that the bias is largely globally consistent, it will not be difficult to apply a correction factor and remove this (systematic) bias'. In reality, it would be difficult to remove systematic bias because "systematic errors are difficult to quantify (Houweling et al., 2014) (P35, L9)."

Thank you for this comment, however we believe these refer to two separate problems. When we reference Houweling with respect to the systematic errors, we are stating that it is difficult to quantify what exactly the systematic errors consist of (spectroscopy or forward model errors etc). So this constant global bias we find, we cannot exactly state what causes this. However if we assume that this bias is constant and global, we can naively apply a correction factor to the whole globe, since all retrievals are effected the same amount by this bias, without direct knowledge of what causes the bias.

Figure 15 shows that the errors in 13CH4 along with aerosol optical depth (AOD) are in Gaussian distribution (no correlation). In reality, there is uncertainty also in AOD. Moreover, other complications come into the retrievals. That is why the systematic error is difficult to be removed. "given that the bias is largely globally consistent" seems too ideal. It would be helpful to mention possible challenges with the real atmosphere.

Thank you for the comment. We agree that retrievals of AOD are also subject to uncertainty, however we have also compared the total uncertainty with the 'true' AOD, based on what was input into the forward model, and we also found no relationship between the uncertainty and the AOD. In this section we are not suggesting that all systematic errors can be removed through a signal global correction, only those caused by the differences of assuming scattering/non-scattering atmospheres with the RemoTeC algorithm.

We have added a clarifying sentence to this point, to identify this point, and highlight that other systematic effects will be more difficult to remove.

Reviewer 2: John Worden

For the purpose of a hopefully more useful review I am identifying myself (John Worden)

Thank you for reviewing our paper.

The paper is generally well written and I understood all the major points and conclusions and consequently I do not have many suggested changes. That said, I do suggest that they try a similar approach to how we estimate the HDO/H2O ratio from TIR radiances (e.g. Worden et al. 2006, Schneider et al., 2012, Lacour et al., 2012) in order to potentially reduce the uncertainties of their delta CH4 retrieval. If such an approach is not feasible in the NIR (although it looks like it should be) then it would be useful to discuss why so that the rationale is documented. Alternatively, add language in the discussion that the TIR based HDO/H2O retrieval approach could be another way for reducing uncertainties in deltaCH4 retrievals.

Just to be sure, from what I understood of their retrieval, the study authors are retrieving the 13CH4 column independently of 12CH4 (although the two are retrieved jointly) such that the constraint for 13CH4 is independent of the 12CH4 retrieval. However, we actually do not scientifically "care" about the 13CH4 column but instead want to know about the ratio of 13CH4 to 12CH4. Our approach for the TIR based HDO/H2O estimate is to therefore jointly estimate the HDO and H2O amounts (similar to the above) but constraining the ratio via cross terms in the constraint matrix. Without this approach the uncertainties in the HDO/H2O ratio were much larger (I think about a factor of 10) than what we report which is about 15-20 per mille (delta-d varies by about 300 per mille so this is an acceptable error). Adding this cross-term is fairly straightforward although I am not yet sure how it fits with the Tikhonov constraint used here but might be as simple as adding a covariance term reflecting prior information about deltaCH4 (e.g. the range is between ~-70 per mille to -20 per mille), see Equation 21 in Worden et al. 2006 "Tropospheric Emission Spectrometer observations of the tropospheric HDO/H2O ratio: Estimation approach and characterization". Without this additional constraint, the errors jump up to ~150 per mille and make the data essentially unusable.

Thank you for this suggestion, we agree that this is certainly something should be investigated. We propose to investigate this in a separate study, since this study is already significant in length. We will add this point to the discussion. We are however concerned that constraining the d13C in such a way would lead mean low information content in the retrievals, and possibly force the retrievals into a profile shape that is not true, as argued by Boesch et al (2013).

Other Comments

The errors is on the order of 1ppb… is that about a 5% uncertainty? (e.g. abundance is ~0.01*2000ppb?). Some language here would be nice as I was initially confused that you could obtain 1 ppb uncertainties when the uncertainty of a total column 12CH4 retrieval is ~10-15 ppb.

We have added additional clarification into the requirements subsection, and the initial errors sections, where we state that we assume the total column VMR of 13CH4 to be 20 ppb, thus a 1 ppb uncertainty of 13CH4 is 5%.

The Jacobians are given as delta Radiance / delta CH4, this was confusing to me at first as delta is also the notation used for the isotopologue, perhaps change to a partial derivative notation instead?

Agreed, done.

publication. Should the editor disagree, and choose to publish the paper as a theoretical exercise in retrieving potential signals larger than those found in the Earth's atmosphere, I have a series of minor comments, outlined below.

These are important points, thank you for raising them. While we cannot replicate the assessment and figures you provided in your review, we are willing to accept the arguments that a 10 per mil requirement is too loose to be of any significant benefit as a measurement, and that our assessment of Fisher's results is not realistic from a total column perspective. You suggest that a precision requirement of 1 per mil is a much more realistic prospect as a useful measurement. We identify in section 3.6 that a precision of 0.02 ppb (required to obtain a d13C precision of 1 per mil) is possible given sufficient temporal and spatial averaging for the SWIR1 retrievals. We therefore propose to update our requirements from 10 to 1 per mil (a precision increase by a factor of 10). We have therefore updated our requirements to reflect this value, and modified all other sections accordingly. We will also post an addendum to our short comment to Professor Roeckmann, updating our arguments.

We discuss how useful this requirement is in the discussion, and emphasis that this study represents a starting point for the science, and that our results warrant further investigation. Significant more work will be required before such retrievals can be made into a data product, or incorporated into a model.

Minor comments

P1, L20: CO2 is not really the "main GHG", but rather the "main anthropogenically-influenced greenhouse gas".

Updated.

A recurring comment: I'm not sure if it's a typesetting issue, but there are several inconsistencies with nested parentheses, including:

P1, L22-23; P2, L8-9; P7, L24-25; P8, L30; P25, L2

Updated

The wrong citation form (again with the parentheses) is used in P3, L19.

Updated.

P3, L32-35: the abbreviation "sect" should be "Sect.", I guess.

Changed

P4, L13: add a comma after citation.

Done

P4, L18: In English colons are usually used for denoting times.

Done

P5, L4: I'd hyphenate it as: "36-layer plane-parallel"

Done

P5, L28: I feel like a word is missing, perhaps "spectral dependence of surface albedo"?

'of' inserted

P6, L14: identified -> described

Couldn't see 'identified', presume 'identify', changed to 'describe'

P6, L21: data "are"; also P9, L22

Changed

P6, L25: The are/will be construction to denote present and future is awkward and should be rewritten.

Modified to:

S5/UVNS will not be launched until 2022 and therefore will not be available until then.

P7, L4: instrumentin -> instrument in

Changed

P8, L28: funny (LaTeX?) quotes on 'truth'; also on P6, L23; P25, L9

Latex related, changed to quotation marks.

P16, L6: should "renders" be "reduces"?

No, renders is correct.

P16, L7: I'd suggest changing "to better knowledge of the ancillary information" to "to improve knowledge of the ancillary information". While "better" can be a verb, here it makes it difficult to parse.

Done

P17, L11: The sentence starting with "While" should be combined with the previous sentence (with a comma).

We could not find the referred to "While", we did find a "Which" in a similar position, and changed following your suggestion.

P21, caption: Should say SWIR3.

Changed

P22, L7-8: Reads a bit awkwardly, could be rewritten.

Changed. Now reads as:

The precision error map shown in Fig. A2 indicates similar levels of systematic error in the SWIR3 when compared to the SWIR1 band.

P23, L2: up roughly -> up to roughly

Changed.

P24, L2: failed convergence -> failed to converge

Changed.

P24, L10: move comma to after "identify"

Done.

P25, L23: comparising -> comparing

Done

P28, L10: makes limited impact -> has a limited impact

Changed

P28, L12: remove comma

Done

P30, L4-5: the Hudson bay -> Hudson Bay

Done

[revised manuscript text omitted]

---

## Author Response (AR3)

Dear Authors,
I am pleased to accept the revised manuscript for publication in AMT subject to a technical change. I propose to acknowledge the critical remarks in the review process that have led to a substantial revision of the stated requirements that would be necessary to distinguish between methane source types.
Best regards,
Andreas Hofzumahaus

Dear Professor Hofzumahaus,

Thank you for the review of our updated manuscript.

We agree with your proposal, and we have inserted the following sentences into the acknowledgements section of our manuscript.

[revised manuscript text omitted]